# More than one landslide per road kilometer – surveying and modeling mass movements along the Rishikesh-Joshimath (NH-7) highway, Uttarakhand, India

Jürgen Mey[1], Ravi Kumar Guntu[2,3], Alexander Plakias[1,4], Igo Silva de Almeida[1], Wolfgang Schwanghart[1]

[1]Institute of Environmental Science and Geography, University of Potsdam, Potsdam-Golm, 14476, Germany
[2]Department of Hydrology, IIT Roorkee, Roorkee, 247667, India
[3]now at GFZ German Research Centre for Geosciences, 14473 Potsdam, Germany
[4]now at Urban Ecosystem Science, Institute of Ecology, Technische Universität Berlin

*Correspondence to*: Jürgen Mey (juemey@uni-potsdam.de)

**Abstract.** The rapidly expanding Himalayan road network connects rural mountainous regions. However, the fragility of the landscape and poor road construction practices lead to frequent mass movements along-side roads. In this study, we
investigate fully or partially road-blocking landslides along the National Highway (NH-7) in Uttarakhand, India, between Rishikesh and Joshimath. Based on an inventory of >300 landslides along the ~250 km long corridor following exceptionally high rainfall during September and October, 2022, we identify the main controls on the spatial occurrence of mass-movement events. Our analysis and modeling approach conceptualizes landslides as a network-attached spatial point pattern. We evaluate different gridded rainfall products and infer the controls on landslide occurrence using Bayesian analysis of an
inhomogeneous Poisson process model. Our results reveal that slope, rainfall amounts, lithology and road widening are the main controls on landslide occurrence. The individual effects of aggregated lithozones are consistent with previous assessments of landslide susceptibilities of rock types in the Himalayas. Our model spatially predicts landslide occurrences and can be adapted for other rainfall scenarios, and thus has potential applications for efficiently allocating efforts for road maintenance. To this end, our results highlight the vulnerability of the Himalayan road network to landslides. Climate
change and increasing exposure along this pilgrimage route will likely exacerbate landslide risk along the NH-7 in the future.

*Keywords:* landslide inventory, road infrastructure, Himalayas, landslide risk, network-attached point pattern

**Statements and Declarations**

*Competing interests:* The authors declare that they have no conflict of interest.

## 1 Introduction

Roads weave across the Earth's surface, with their number and extent continually expanding. Compared to 2010, the total length of roads globally is expected to enhance by 60% until 2050 (Laurance et al. 2014), increasingly dissecting and

fragmenting ecosystems. In mountainous regions, road construction leads to profound changes in the stress distribution of adjacent hillslopes, redistribution of rock and soil, and alterations of drainage patterns. If poorly implemented, road construction, inapt slope protection and insufficient drainage thus enhance slope instability and the frequency of landslides, with substantial economic damages and increasing number of fatalities (Bíl et al., 2014, Laimer 2017, McAdoo et al., 2018, Muenchow et al., 2012, Ozturk et al., 2022). A worldwide surge in the proposals for new roads thus warrants the investigation of the link between road network expansion and landslide risk (Laurance et al., 2014).

Roads are at the heart of the Himalayan transport infrastructure. They are vital for national and international trade and passenger movement and strategically important in border areas. India has improved and expanded its road network in mountainous states, e.g. under the national Bharatmala Pariyojana ("Road to Prosperity") initiative, established in 2015. Key objectives of this highway development program are to improve the efficiency and connectivity of the transport infrastructure and to provide road access to remote border regions and rural areas (Boora & Karakunnel, 2024). Yet, in mountainous environments, roads are exposed to various degrading processes. Among these processes, mass movements particularly inflict severe structural damage and heavily degrade road serviceability (Meyer et al., 2015). Traffic disruptions due to mass movements can have severe consequences, if they impede accessibility and compromise rescue operations during extreme events such as cloudbursts, floods and earthquakes (Sharma et al., 2014). Ensuring accessibility and connectivity during such events is thus of live-saving importance, yet requiring considerable maintenance efforts (Uniyal, 2021).

According to the National Crime Records Bureau (2022), 160 people died due to landsliding in Uttarakhand in the preceding four years. These figures exclude other extreme events like heavy rainfalls, floods or the 2021 Chamoli rock and ice avalanche with over 200 fatalities (Shugar et al., 2021). Several studies have addressed mass movements and their relation to transport infrastructure in the Indian and Nepal Himalayas. The studies range from purely phenomenological descriptions (e.g., Bartarya & Valdiya, 1989; Sarkar & Kanungo, 2006), to statistical (Das et al., 2012; Devkota et al., 2013; Sur et al., 2020) and physically based modeling approaches (e.g., Kanungo et al., 2013; Prasad & Siddique, 2020). In fact, the space limitation in steep terrain often requires road construction to undercut slopes beyond their angle of repose, reducing slope stability and increasing landslide susceptibility (e.g., Barnard et al., 2001; Haigh & Rawat, 2011; Li et al., 2020). The increase in mass wasting due to road construction itself may be similar to the impact of a large earthquake (Tanyas et al. 2022). Therefore, particular attention has focused on detailed stability assessments of road cut slopes (e.g., Kundu et al., 2016; Siddique et al., 2017; Siddique & Pradhan, 2018; Singh et al., 2014) and the development of appropriate remedial measures (e.g., Adhikari et al., 2020; Asthana & Khare, 2022; Koushik et al., 2016; Rawat et al., 2016), but fewer studies have attempted to predict the spatial occurrence of mass movements along roads (Huat & Jamaludin, 2005; Ching et al., 2011). Knowledge about where and when landslides preferably detach is important for early warning but also for efficiently allocating efforts of road maintenance and slope enforcement (Haigh, 1984). Using data on occurrences of landslides, susceptibility studies aim to quantify the spatial propensity of hillslopes to fail and to determine the controlling factors such as terrain attributes e.g., slope angle and aspect, and geo-environmental variables e.g., rainfall intensity and lithology.

In this study we carry out an analysis of road exposure to landsliding for a ~250 km long stretch of the National Highway 7 (NH-7) that connects the cities of Rishikesh and Joshimath, Uttarakhand, India (Fig.1). We conducted a detailed survey of partially or fully road-blocking landslides along the road, following the monsoon season (May – Aug) and a period of intense rainfalls during 5 - 12 October 2022. Limiting the inventory to road blocking landslides was required to cope with the overwhelming number of landslides and to ensure that we account for the landslides that detached most recently. In addition, construction works for road widening often stripped away the vegetation so that differentiating between actual, non-blocking

landslides and excavated slopes was not straightforward. In contrast to previous studies, which focused on the prediction of landslides in two spatial dimensions (e.g., Das et al., 2012; Devkota et al., 2013), our analysis and modeling approach conceptualizes landslides as a network-attached spatial point pattern (Baddeley et al., 2021).

80

Landslide susceptibility analysis commonly relies on discretizing the study region into pixels, where each pixel indicates either the presence or absence of landslides. Subsequently, techniques such as logistic regression are used to predict the probability of a landslide as a function of a number of predictor variables. A pixel-based logistic regression analysis is approximately equivalent to a Poisson point process model if the landslides are originally stored as point features (Baddeley et al. 2010). We take this approach to modeling landslide susceptibility using point process models a step further by conceptualizing landslides as point features that are located on or along-side the road network (Okabe and Sugihara 2012). This means that we do not analyze road-attached landslides as events that occur on a continuous and unbounded plane, but rather as events that are along-side of roads. The term "along-side" indicates a somewhat broad spatial relation, but "implies that the physical unit of the event [...] has an access point on a network" (Okabe and Sugihara 2012, page 7). In our case, this means that a landslide intersects with the road and blocks it. These intersections are, at the scale of our analysis, represented by point features, i.e. points on a network. Our approach relies on the analysis of the spatial arrangement of the points (the point pattern) along the line, and we hope to reveal important features (e.g. trends in point density) with respect to geo-environmental variables. An additional objective of this study is to test, whether such an approach would be particularly feasible to determine the inter-relationship between linear infrastructure and landslides. One of the critical covariates in our modeling approach is the spatial distribution of accumulated rainfall amounts. We thus evaluate different rainfall products. Finally, we infer the controls on landslide occurrence using Bayesian analysis of multivariate loglinear models.

## 2 Study area

The NH-7 ascends from 400 m at Rishikesh to approximately 2000 m at Joshimath, crossing steep terrain with soil mantled slopes. Mean annual rainfall (1970–2019) varies from 1500–2000 mm around Rishikesh to 1000–1200 mm in Joshimath with 80–86 % and 60–70 % delivered by the Indian summer monsoon (June to September), respectively (Pai et al., 2014; Swarnkar et al., 2021). Air temperature in Rishikesh is always above freezing and ranges between 4 and 40 °C, whereas the temperature in Joshimath varies between -10 and 20 °C. This climatic gradient is evident in the gradual transition of vegetation. In the lower lying subtropical region, dense deciduous forests dominate. As elevation increases, these forests give way to temperate broadleaf mixed forests and temperate shrub and grassland communities.

105

The geological framework of the study area is largely determined by the ongoing Indo–Asian collision that causes crustal thickening and exhumation along large-scale detachment zones and thrust faults. Most of the study area lies within the Lesser Himalaya, between the Main Boundary Thrust in the south and the Main Central Thrust (MCT) in the north, which are both splays of the root detachment, the Main Himalayan Thrust (Fig. 1). As the present-day India–Eurasia convergence is on the order of 36–40 mm yr$^{-1}$ (e.g., Wang et al., 2001) and approximately half of this is accommodated within the Himalayas (e.g., Lavé & Avouac, 2000), the region is seismically active and bears the potential for large earthquakes (e.g., Kayal et al., 2003; Bollinger et al., 2014; Rajendran et al., 2017).

The highway runs perpendicular to the strike of the orogen and crosses rocks of the Lesser Himalayan Sequence (LHS) and the High Himalayan Crystalline (HHC). These units represent the ancient passive Indian margin and are separated by the MCT. The LHS is mainly composed of sedimentary and low-grade metasedimentary rocks; quartzite, shale, phyllites and slate with occasional limestone and dolomite, whereas the HHC is characterized by high-grade schist, gneiss and quartzite.

These rocks feature a high density of discontinuities like faults, fractures and joints that are important seepage pathways. In locations, where the road cuts through weathered rocks and intersects with major faults, the hillslopes are particularly fragile (Prasad & Siddique, 2020).

During the week preceding our survey, large parts of North India experienced a period of strong rainfall. The state of Uttarakhand registered a departure of 1040 % from the long-term (1961 – 2010) average. The districts of Tehri Garhwal, Pauri Garhwal, Rudraprayag and Chamoli, through which the NH-7 runs, recorded a weekly surplus of 419 %, 679 %, 218 % and 1855 %, respectively (supplementary table 2). Given that the average rainfall amount in October is around 35 mm, approximately three times the monthly average rainfall occurred in only one week, which is close to the rate that prevails during the monsoon.

The NH-7 is a lifeline for socioeconomic development, which is mainly based on agriculture, trade, tourism, mining and hydropower. Furthermore, the highway is vital for the Indian military to transport personnel and equipment to their outposts along the Indian/Chinese-Tibetan border. Between May and October, more than one million pilgrims travel the highway to visit the holy shrines of Badrinath and Kedarnath. Moreover, the highway follows the course of the Ganges and its tributary Alaknanda and thus passes the river confluences known as the five Prayags, namely Devprayag, Rudraprayag, Karnaprayag, Nandaprayag and Vishnuprayag (Fig. 1). In Hinduism, these confluences are considered sacred, attracting pilgrims who bathe in the waters before worshiping the rivers. Finally, there are ten hydroelectric power plants within 20 km distance to the road and more projects are planned or under construction, highlighting the road's significance for energy security and economic value.

## 3 Data and Methods

Traveling to fieldwork in the Chamoli area (Uttarakhand) on Oct 15, 2022, we recognized numerous, partially road-blocking landslides along the road and decided to investigate their occurrence using the following approach.

First, we started to record these landslides using hand-held GPS devices as road workers already began to clean the road from the debris, thus rapidly removing evidence of smaller landslides that had detached in close vicinity to the road. We registered landslides along the road both on our way from Rishikesh to Joshimath (Fig. 1) on Oct 15 and 16, as well as on our way back on Oct 18, 2022. We only recorded landslides with runouts affecting the road, thus partially or fully blocking it (Fig. 2). We classified the landslides as partially-blocking if the emplaced deposits either substantially narrowed the road, or if debris ran over marginal road markings. Very small landslides with an area of less than ~10 m$^2$ were not considered. Secondly, we cross-checked each recorded landslide using Google Earth using most recent and historic high-resolution imagery. The images enabled us to identify the landslides that occurred between January and October 2022, encompassing the entire, annual monsoon season as well as the heavy rainfall period at the beginning of October 2022. We initially tried to attribute the landslides to the latter period by matching our data with acquisition dates of the images, but the temporal resolution for many road sections was insufficient. Moreover, historic imagery has become unavailable for India (https://www.thehindu.com/news/national/google-historical-satellite-imagery-disappears-for-india/article66834033.ece, last access: 25 June 2024). We also identified reactivated landslides, where a slip surface and a scar were visible in the imagery prior to Jan 2022. This was not always straight-forward since landslide scars cannot always be clearly distinguished from bare, engineered slopes and road widening. Landslides confidently determined to have occurred before January 2022, as well as those obscured by shadows in Google Earth images, were omitted from the analysis.

Thirdly, we analyzed landslides along the road as a network-attached spatial point process. A spatial point process is a stochastic mechanism, which controls the spatial distribution of events or occurrences (Baddeley et al., 2015). As our mapped landslides are events that occur along the road – and only these have been mapped – these events are constrained to lie on a network of lines (Baddeley et al. 2021, Okabe et al. 2006). In our case, the network is rather simple as it consists of only one polyline, but the approach can be equally applied to more complicated network topologies.

We used the open-source MATLAB extension TopoToolbox (Schwanghart and Scherler, 2014) and its numerical object PPS (Schwanghart et al., 2021) to analyze, visualize, and model the density of landslides along the road. PPS means "Point patterns on stream networks", but it can be used with any type of dendritic, undirected network. The numerical approach consists of a fine-pixel approximation which is controlled by the geometry of the digital elevation model (DEM), from which the data is retrieved. This means that the vector shape of the road is pixelated with the same geometry as the DEM (Schwanghart et al., 2021). We model landslide densities with an inhomogeneous Poisson process model, which is defined by the intensity function $\lambda(u)$ with $u$ being the horizontal distance along the road. A common parametric model of the intensity is the loglinear model:

$$\lambda(u) = exp(\beta_0 + \beta X(u)),\qquad\qquad\qquad\qquad(1)$$

where $X$ is a matrix of predictor variables (covariates) that vary along the road, $\beta_0$ is an intercept and $\beta$ is a vector of model parameters. Our approach uses a spatial logistic regression, where the relation between presence probabilities $p$ and explanatory variables $X$ are controlled by the form of the logistic link function (logit $p = \ln(p/(1-p))$). As pixel size tends to zero, we have $p \rightarrow 0$ and $\ln(p/(1-p)) \rightarrow \ln(p)$. The limiting Poisson intensity is thus a loglinear function of the covariates (Baddeley et al. 2010). A key property of the model is that the events are independent from each other, i.e. the probability of a landslide to occur is independent from landslides nearby. Spatial dependence of events can occur in different ways leading to clustering, i.e., points tend to occur close to other points, or inhibition, i.e., there is a characteristic distance or regularity in the spacing between the points. Spatial clustering of landslide events has previously been addressed by Lombardo et al. (2018, 2019) using a Cox Process model to emulate the latent spatial effects of unobserved variables, whereas inhibition can be observed, for example, in data where areal non-overlapping phenomena are represented as points (Evans, 2012; Schwanghart et al., 2021). At this stage, we will not include these potential second-order effects on the density of landslides in our model, but we will investigate their possibility using the inhomogeneous K-function defined by Ang et al., (2012) once we have modeled first-order effects.

We used the following candidate predictor variables in the loglinear model introduced above. First, we hypothesized that steep hillslopes gradients next to the road are more susceptible to mass wasting events. Therefore, we calculated surface gradients, based on the Copernicus 30-m DEM (European Space Agency, 2021). We identified areas that lie right or left to the road, and which are higher than the road itself within a buffer zone of ~210 m (or 7 pixels). We then selected the nearest DEM pixels and mapped the mean slopes of these nearest pixels to the road network. We chose 210 m (7 x spatial resolution of the DEM) to have sufficient pixels to obtain robust estimates of the slope. These values still vary greatly over short distances along the road and thus we smoothed them using the algorithms (with smoothness penalty parameter K = 4) described by Schwanghart and Scherler (2017).

Rainfall patterns exert a strong influence on the spatial occurrence of landslides (e.g. Ching et al., 2011; Joshi & Kumar, 2006; Gariano & Guzzetti, 2016, Ozturk et al., 2021). Thus, we chose five rainfall/precipitation products (see Table 1) to characterize spatial patterns of accumulated rainfall between January 1 and October 10, 2022. Among the five rainfall/precipitation products, IMD1 solely relies on the interpolation of gauge-based measurements from a network of

stations provided by the Indian Meteorological Department (Pai et al., 2014). In addition to gauge measurements, IMD2 is enhanced with estimates from the Integrated Multi-satellite Retrievals for Global Precipitation Measurement (IMERG) (Mitra et al., 2009). MSWEP v2 is another merged product that incorporates reanalysis-based, gauge, and satellite-derived rainfall estimates (Beck et al., 2017). CHIRPS v2 provides a high-resolution record by combining gauge and satellite data from NOAA (National Oceanic and Atmospheric Administration) (Funk et al., 2015) and the IMERG late run provides gridded multi-satellite precipitation estimates with quasi-Lagrangian time interpolation from NASA (National Aeronautics and Space Administration) (Huffman et al., 2014). We projected the geographic coordinates of the rainfall/precipitation grids to UTM zone 44N and bilinearly resampled the data to the resolution of the DEM before extracting the values of each road pixel.

Next, we incorporated information about road widening by creating a logical mask that identifies the widened segments of the highway. Due to the absence of more detailed data, we rely on the map by Sati et al. (2011), which shows the locations of road widening completed before the 2010 monsoon season. Our envisioned strategy of compiling a more recent database of road widening using historic imagery from Google Earth was rendered impossible after most of the historic images were removed from the public archive in April 2023.

Finally, we obtained a digitized version of the lithological map of Uttarakhand (map scale 1:2.000.000) from the Geological Survey of India (2022). The data contains both stratigraphic as well as lithological information. Accordingly, the NH-7 crosses 34 different lithologies between Rishikesh and Joshimath. To reduce the number of categories, we summarized and aggregated the lithological information into lithozones with lesser focus on the stratigraphic context. This aggregation resulted in five lithozones that are dominated by carbonate rocks (1), phyllite and shale (2), quartzite (3), quartzite and igneous rocks (4) and crystalline high grade metamorphic rocks (5). The reclassification is shown in Table 2. Again, we gridded this data and assigned corresponding lithozones to each road pixel.

We adopt a Bayesian strategy to infer and identify predictor variables using the function bayesloglinear of the PPS numerical class (Schwanghart et al., 2021). The function provides an interface to bayesreg (Makalic and Schmidt, 2016), a MATLAB toolbox that enables efficient Bayesian modeling and regularization of high-dimensional data. We use a Bayesian lasso estimator with Laplace prior distributions for the regression coefficients (Park & Casella, 2008). This prior results in posterior mode estimates that are similar to estimates obtained with the lasso penalty (van Erp et al., 2019). Before calculating 1000 samples of the posterior parameter distributions, we calculated 1000 burnin samples. These are calculated to ensure that the Gibbs sampler converges to the target distribution. In addition, we used a level of thinning of five samples. This means that only every fifth sample was retained in the generated sequence to reduce autocorrelation between the samples and to obtain a more independent and representative Bayesian posterior distributions. To this end, we find that 1000 samples are sufficient to characterize the posterior distributions.

Finally, we evaluate the model based on the Receiver-Operating Characteristics (ROC) Area under the Curve (AUC) approach (e.g., Hanley and McNeil, 1982). We visualize the predictions and inspect and analyze spatial densities obtained from random realizations of the fitted inhomogeneous Poisson process model. Moreover, we evaluate the predictive performance of the model using a 5-fold cross-validation. In this approach, we subdivide the road into 15 km segments, which are then randomly partitioned into 5 groups. The first group is used as test data while the remaining groups are used to train the model. This step is then repeated for each group and the performance summarized with the AUC. In addition, we test whether additional covariates provide opportunities for further improving the model. The selected attributes include terrain roughness and total curvature as well as land cover derived from the Copernicus Global Land Operations (CGLOPS-

1, Moderate dynamic land cover 100 m, version 3) (Buchhorn et al., 2020), which we reclassify according to Table 3. We investigate these models using a frequentist modeling approach (see PPS-function fitloglinear) and compare models with additional covariates with the Akaike Information Criterion (AIC, Akaike, 1974).

## 4 Results

We recorded 309 fully or partially road-blocking landslides along the 247 km long road between Rishikesh and Joshimath, which amounts to an average landslide density of 1.25 landslides per km. A two-sample Kolmogorov-Smirnoff test between the road distance (uniform distribution between start and end of the surveyed road), and the road distances measured at the landslides rejects the null hypothesis with p ≈ 0 that landslide locations follow a completely spatial random distribution. Visually inspecting the landslide locations using Google Earth reveals that 80 % of the recorded landslides with road blockages occurred after Jan 2022 (Fig. 1). Of these 250 landslides, 30 % were most likely reactivated because they could not be identified to be road-blocking before the rainy season (Fig. 2e,f).

The spatial distribution and amounts of accumulated rainfall between January 1 and October 10 differ between the rainfall products (Fig. 3). Since independent measurements based on rain gauges are unavailable, we investigate the performance of the rainfall products to explain the spatial distribution of landslides. The approach uses the AIC to iteratively evaluate models including one of the rainfall products at a time, as well as hillslope gradient, lithozones, and road widening. AIC values vary between 1876 and 1913 with CHIRPS v2 having the lowest AIC. CHIRPS v2 correlates positively with landslide density, whereas all other rainfall products show negative or no significant correlation (see supplementary table 1 and supplementary figures 1-5). We thus use CHIRPS v2 in the development of the subsequent models. We emphasize that including different rainfall products in the model has no strong effect on the remaining model parameters that determine the influence of slope and lithozones and road widening (see supplementary table 1). In other words, while determining the overall performance the choice of rainfall product does not affect our results and conclusions about the topographic and lithologic controls on landslide occurrence.

Bayesian loglinear modeling of the landslide density (Fig. 4, Fig. 5a, b) reveals a credible influence of the covariates rainfall (Fig. 5c), slope (Fig. 5d), lithozones (Fig. 5e) and road widening (Fig. 5f) (see Fig. 4 for posterior means and 95 % highest density intervals and Fig. 6 for individual effects). A Bayesian feature rank algorithm based on the absolute magnitude of the parameters in each posterior sample (Makalic and Schmidt, 2011), ranks slope as the top covariate in terms of explanatory power together with rainfall, followed by road widening and lithozones (Table 4). Among the lithozones, zones 2 and 4 stand out as categories improving the explanatory power of the model. The individual effects of the covariates reveal a positive influence of rainfall and slope on landslides (Fig. 6a, b). Predictions of landslide densities in lithozone 4 are credibly lower than in lithozone 2 (Fig. 6c). Also, our model suggests that road widening affects the density of landslides. Corrected for the influence of other parameters, average densities are twice as high in widened road sections (Fig. 6d). The spatial pattern of predicted landslide density (Fig. 5g) is consistent with observed spatial density variations, but the high small-scale variability reflects the importance of slope as a predictor variable.

The AUC is an aggregated metric for a point pattern model across thresholds and ranges between 0.5 and 1, with values close to 1 indicating high performance of a model. Our loglinear model has an AUC value of 0.79 (Fig. 7a). The inhomogeneous K-function shown in Fig. 7b quantifies the expected number of points as a function of distance from each point, adjusted for the modeled inhomogeneous intensity of the point pattern. Distances between individual landslides are calculated as the shortest-path distance along the road rather than the direct Euclidean distance. Acceptance intervals around

the theoretical K-function were derived from repeated simulations of the inhomogeneous Bayesian loglinear model. The actual point pattern's K-function is within these acceptance intervals, suggesting that the landslide locations do not exhibit clustering. A comparison of 100 randomly simulated and actual point densities (Fig. 7c) shows that the modeled and observed spatial landslide densities are consistent although the second, smaller peak of landslide densities close to Joshimath are not well captured by the model. Moreover, 5-fold cross-validation based on 15-km long road segments reveals AUC-values between 0.77 and 0.78.

Can the model be improved by incorporating more explanatory covariates? Our impression in the field was that landslides detach independently of planform hillslope geometry as they occurred both on spurs and convex hollows. Nevertheless, we calculated total curvature and topographic roughness as potential predictor candidates. In addition, we used landcover (Table 3) and distance to faults (Fig. 1) as they are commonly used in susceptibility studies (e.g., Stanley and Kirschbaum 2017, Li et al., 2020, Ozturk et al., 2021) and potentially control the density of road-side slope failures. Yet, including these metrics in the model barely contributes to improving the model fit and their incorporation in the model would, according to the constancy of the AIC, lead to overfitting (Fig. 8).

## 5 Discussion

We recorded more than one landslide per road kilometer along the NH-7 between Rishikesh and Joshimath. The fact that this road is strongly affected by landslides has been previously described and attributed to the region's fragile slopes, intense rainfall and frequent seismicity (Sati et al., 2011). In addition to the environmental conditions, road construction and widening have contributed to the occurrence of new landslides, which are often shallow and small, but nevertheless lead to fatalities, inflict severe damages to infrastructure and cause traffic disruption (Sati et al., 2011). We conducted a systematic survey of road-side landslides and derived a statistical model that quantifies landslide susceptibility along the NH-7 at a high spatial resolution.

Our analysis relied on a GPS-based survey of landslides while traveling from Rishikesh to Joshimath shortly after a period of anomalously high rainfall. Using this approach, we mapped landslides irrespective of cloud cover and without acquiring high-resolution satellite imagery, which is usually needed to reliably detect small landslides. A drawback, however, is that we may have missed landslides where debris had already been removed by road works. Also, detailed mapping of the areal extent of the landslides was not possible during drive-by so that we did not quantify the size of landslides. Thus, our analysis treats all landslides the same, irrespective of their areal extent and volume. To this end, however, this enables us to adopt a modeling approach, which conceptualizes landslides as an unmarked, network-attached point pattern (Baddeley et al., 2021; Okabe and Sugihara, 2012). Representing landslides as network-attached points and not as areal features entails advantages and disadvantages. Constraining landslide locations to lie on roads demands that all predictor variables also need to be mapped to the road network, which entails some generalization and additional degrees of freedom about the choice and aggregation of 2D variables. For landslide susceptibility analysis, for example, this means that spatial variables characterizing the source area (e.g., hillslope gradient) are projected onto the road. At the same time, model development and fitting benefits from smaller sample sizes as data amounts are moderate and computational demands during Bayesian posterior sampling remain manageable.

We detected a profound difference between rainfall products, for which a detailed analysis is outside the scope of this study. Several studies along the Himalayan orographic front have previously detected these differences. They have been attributed to the sparse network of rain gauges, or, in cases where rainfall estimates are based on remote sensing data, to irregular

acquisition times, which can result in missing individual rainfall events (Andermann et al., 2011, Hu et al., 2016). To this end, these uncertainties are detrimental to accurately capturing the spatial patterns of landslides (Ozturk et al., 2021). We found that CHIRPS v2 performed best in predicting the spatial landslide patterns along NH-7 but the employed search strategy must be viewed critically as the reverse conclusion, that landslides are controlled by these patterns is not necessarily true. Indeed, studies come to different conclusions about the performance of CHIRPS v2 and other gridded rainfall products. For example, Kumar et al. (2021) studied different gridded rainfall products in the Eastern Himalayas and found that CHIRPS-v2 overestimated the monsoon but underestimated annual precipitation. Based on the analysis of 18 extreme precipitation events during 2014–16 in the northwest Himalaya (including our study site), however, Jena et al. (2020) concluded that CHIRPS v2 provides the most reliable precipitation estimates. Indeed, the two-peaked rainfall pattern of CHIRPS v2 is most consistent with long-term rainfall patterns obtained from the interpolation of 44 rainfall gauge records averaging over the period from 1901 to 1950, which show highest values along the Himalayan Front and the physiographic transition to the High Himalaya (Basistha et al., 2008; Bookhagen, 2010).

Lithozones derived from a geological map contributed to the explanatory power of the model, thus highlighting the role of rock properties in modulating landslide susceptibility. As we did not measure the actual geotechnical and -mechanical properties or e.g., bedding and foliation along our route, we can only provide a first order reasoning of the prediction capacity of the lithozones. The high landslide density in lithozone 2 is likely related to the pronounced fissility and cleavage of the dominating shales and phyllites. Moreover, material softening, percolation and weathering cause a general decrease in rock strength. Tectonic activity adds to a general decrease in rock strength by creating shear surfaces with low friction angles (Stead, 2016). In addition, road segments, where the adjoining hillslopes parallel bedding, joints or foliation planes are particularly vulnerable (e.g., Bartarya and Valdiya, 1989). Conversely, lithozone 4 is characterized by quartzite and igneous rocks. These have undergone low- to high-grade metamorphism and are generally harder and have a more irregular fabric that restrains the formation of planar slide surfaces. Moreover, these rocks tend to develop more stable regolith mantles (Gerrard, 1994), and are thus less susceptible to landsliding. Our model shows that under the same topographic conditions and rainfall amounts, the rock types of lithozone 2 are 2–6 times more susceptible to landslides than those in lithozone 4 (Fig. 6c). The remaining lithozones are not credibly different from each other. In part, the lack of differences can be attributed to the low number of samples as only a small fraction of the road traverses these lithozones (Table 2). Notwithstanding, a general trend towards lower landslide susceptibility from lithozone 1–5 are consistent with a previous review study about lithological controls on the occurrence of mass movements in the Himalaya (Gerrard, 1994).

Previous studies indicate that human activities have played a crucial role in predisposing slopes to failure (Li et al., 2020; van Westen et al., 2008; Tanyas et al., 2022). Our analysis underscores these findings and quantifies the occurrence of landslides to be twice as high along widened sections of NH-7, if other predictor variables are held constant. The process of cutting into mountain sides to create wider pathways often creates unstable slopes that are prone to failure (Barnard et al., 2001; Haigh & Rawat, 2011; Sati et al., 2011; Li et al., 2020). Clearing trees and vegetation for road widening eliminates their stabilizing influence on slopes, thereby increasing landslide hazard (e.g., Guthrie, 2002). Additionally, widened roads can lead to increased surface runoff during heavy rainfall or snowmelt, saturating the soil and making it more susceptible to landslides (e.g., Wadhawan et al., 2020). Rock blasting required during the road widening process can lead to the fracturing and weakening of rock masses, creating potential landslide-prone zones along the road corridors (Sati et al., 2011). Moreover, road widening alters natural drainage patterns, and potentially redirects water flow to adjacent slopes, thus causing water saturation, erosion and instability. In fact, these disturbances have previously led to frequent landslides along the NH-7 as Sati et al. (2011) also report about ~300 landslides occurring along the road more than 10 years ago. Our data indicates that 30 % of the recorded landslides are reactivated slope failures which highlights that slopes are recurrently

unstable during periods with intense rainfall (Joshi and Kumar, 2006). During mapping, we also noticed that some slopes were engineered during the last years with retaining walls, yet many of which also failed.

Clearly, our model may miss important predictor variables that control the occurrence of landslides. We included variables that characterize environmental conditions and found that slope, rainfall, road widening and lithology largely explain the variability in landslide density. Variables such as landuse or topographic derivatives do not improve the performance of the model as measured by the AIC, at least at the spatial scale at which these variables were available. However, small-scale topographic changes due to construction or land use changes (e.g. abandonment and degradation of agricultural fields and terrace systems) may exacerbate road-side slope failures (Jacquet et al., 2015; Mauri et al., 2022). A more up-to-date DEM with higher resolution may indeed help to improve the spatial prediction of landslide, although higher DTM resolutions were shown to not necessarily improve model performance, in particular along roads (Penna et al., 2014). Moreover, the propensity of landsliding along widened segments of the road may change over time. Road widening and slope undercutting can be viewed as a perturbation, to which slopes will adjust. Timescales of this adjustment may vary and depend on several factors. Likewise, remedial measures such as slope enforcement, artificial drainage and retention walls will affect slope stability. Including these activities of landslide mitigation was not possible in our study, but could support planning of structural measures.

Our model hindcasts the spatial pattern of road-blocking landslides and we posit that it can be used as a time-predictive model as well. Rainfall is one of the main covariates in the model and is also the one with the largest uncertainties as shown by the discrepancies of gridded rainfall products. A denser network of rain gauges and a better availability of this data would likely contribute, together with weather forecasting, to more accurate estimates of landslide occurrences, which ultimately would facilitate a more efficient allocation of resources for road maintenance. Also, recurrent slope failures should be monitored more closely to direct efforts for slope reinforcements. The land cover data, which we included in the model, is too coarse to capture the widespread lack of vegetation along the road. As many of the landslides were shallow, revegetating slopes may contribute to their stabilization (Vergani et al., 2017).

The NH-7 is a key arterial road and landslides make transport of goods and people difficult, thus causing serious economic disruption. Moreover, slope failures along the road have led to fatalities in parts where roads were widened, and recent heavy rainfalls in July 2023 have caused several severe landslides ([https://www.hindustantimes.com/india-news/landslide-on-char-dham-road-leaves-pilgrims-stranded-heavy-rains-cause-damage-and-blockages-in-uttarakhand-101690225215947.html](https://www.hindustantimes.com/india-news/landslide-on-char-dham-road-leaves-pilgrims-stranded-heavy-rains-cause-damage-and-blockages-in-uttarakhand-101690225215947.html), last access: 26 June 2024). Such damages and commensurate fatalities may become even more frequent in the future. The entire Upper Ganga basin is susceptible to extreme rainfall events (Joshi and Kumar, 2006), and climate change projections – although subject to high uncertainties – indicate a trend towards more frequent extreme events due to elevation-dependent warming and a likely increase of summer monsoon precipitation by 4–25 % (Krishnan et al., 2019). In addition, exposure to landslides is likely to increase in Uttarakhand as the Char Dham National Highway project gets implemented (Chouhan et al., 2022). Road construction and increased traffic volumes attract more people, who will strive for new economic opportunities associated with sites along roads (Fort et al., 2010; Chouhan et al., 2022). These sites are often more susceptible to landslides as construction often implies vegetation removal and slope destabilization (Petley et al., 2007, Li et al., 2020). A reduction of traffic may disrupt the cycle of increasing hazard and exposure. The commissioning of the currently constructed 125 km long broad-gauge railway between Rishikesh and Karnprayag (Azad et al., 2022) might be a major step towards this goal.

## 6 Conclusions

Road construction is soaring in the Himalayas. During the last five years, ~11,000 km of roads were built in the Indian Himalayan states (The Tribune, 2022). Yet, the fragility of the Himalayan landscape as well as slope undercutting and poor construction practices make maintenance of these roads challenging. Our study of landslides along the NH-7 demonstrates the scale of this challenge as we detect more than one partially or fully road-blocking landslide per road kilometer between Rishikesh and Joshimath. We contribute to a better understanding and prediction of these landslides by creating a landslide inventory of road-side landslides and the adoption of a novel approach to landslide exposure analysis, which treats the landslides as unmarked network-attached spatial point phenomena. Together with inhomogeneous Poisson process models, this inventory enables us to identify the main controlling variables, i.e., slope angle, rainfall amount, road widening and lithology. Our model shows that, if corrected for all other influences, road widening leads to a doubling of road-blocking landslides. This finding quantitatively underpins previous claims that improper construction practices of road widening increase, rather than decrease landslide hazards along roads (Sati et al., 2011). The Himalaya's fragile geology and exposure to torrential rains demand proper geological assessments and mitigation measures, such as constructing retaining walls, installing drainage systems, and stabilizing slopes, to minimize the impact of road widening on landslide occurrence. Thorough environmental impact assessments before road widening projects and consideration of alternative routes or transportation solutions can help strike a balance between development and preserving the delicate mountain ecosystem while reducing landslide risks.

*Code and data availability.* The code and data to run the analysis are available in the supplemental data.

*Author contributions.* All authors conceived the study and conducted mapping landslides. RKG retrieved and analyzed the rainfall data. JM and AP compiled, analyzed and visualized the data and WS conducted the statistical analysis. All authors wrote the manuscript.

*Acknowledgments.* We acknowledge the German Exchange Service (DAAD) for funding the project Co-PREPARE (Project-ID: 57553291), a joint project of the IIT Roorkee and University of Potsdam. JM acknowledges financial support from the Research Focus "Earth and Environmental Systems" of the University of Potsdam. RKG acknowledges the financial support received from the Prime Minister's Research Fellowship, Government of India for Grant number PM-31-22-695-414.

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

**Figures**

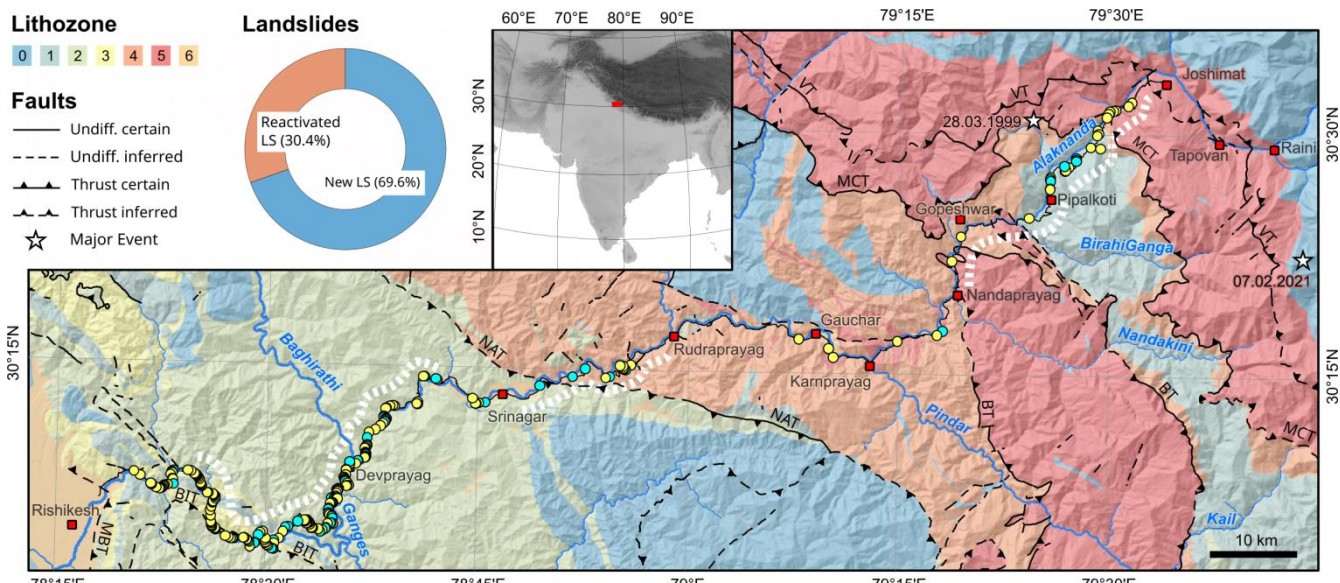

**Figure 1: Map of the study site.** Landslides, lithozones and major faults along NH-7 from Rishikesh to Joshimath. Lithozones were defined according to their dominant lithology: (1) carbonate rocks; (2) phyllite and shale; (3) quartzite; (4) quartzite and igneous rocks; (5) crystalline high grade metamorphic rocks (Table 2). Note that lithozones 0 and 6 are not
crossed by the road and are therefore omitted from the description. We subdivided the landslides into new ones and reactivated ones. White stippled lines indicate the road segments that have undergone widening (Sati et al., 2011). Lithozones and faults were modified from digital maps provided by the Geological Survey of India (2022). Stars indicate locations of the 1999 $M_w$ 6.6 Chamoli earthquake (Kayal et al., 2003; USGS, 2022) and the 2021 Chamoli rock and ice avalanche (Shugar et al., 2021). MBT: Main Boundary Thrust, BIT: Bijni Thrust, NAT: North Almora Thrust, BT: Baijnat
Thrust, MCT: Main Central Thrust, VT: Vaikrita Thrust.

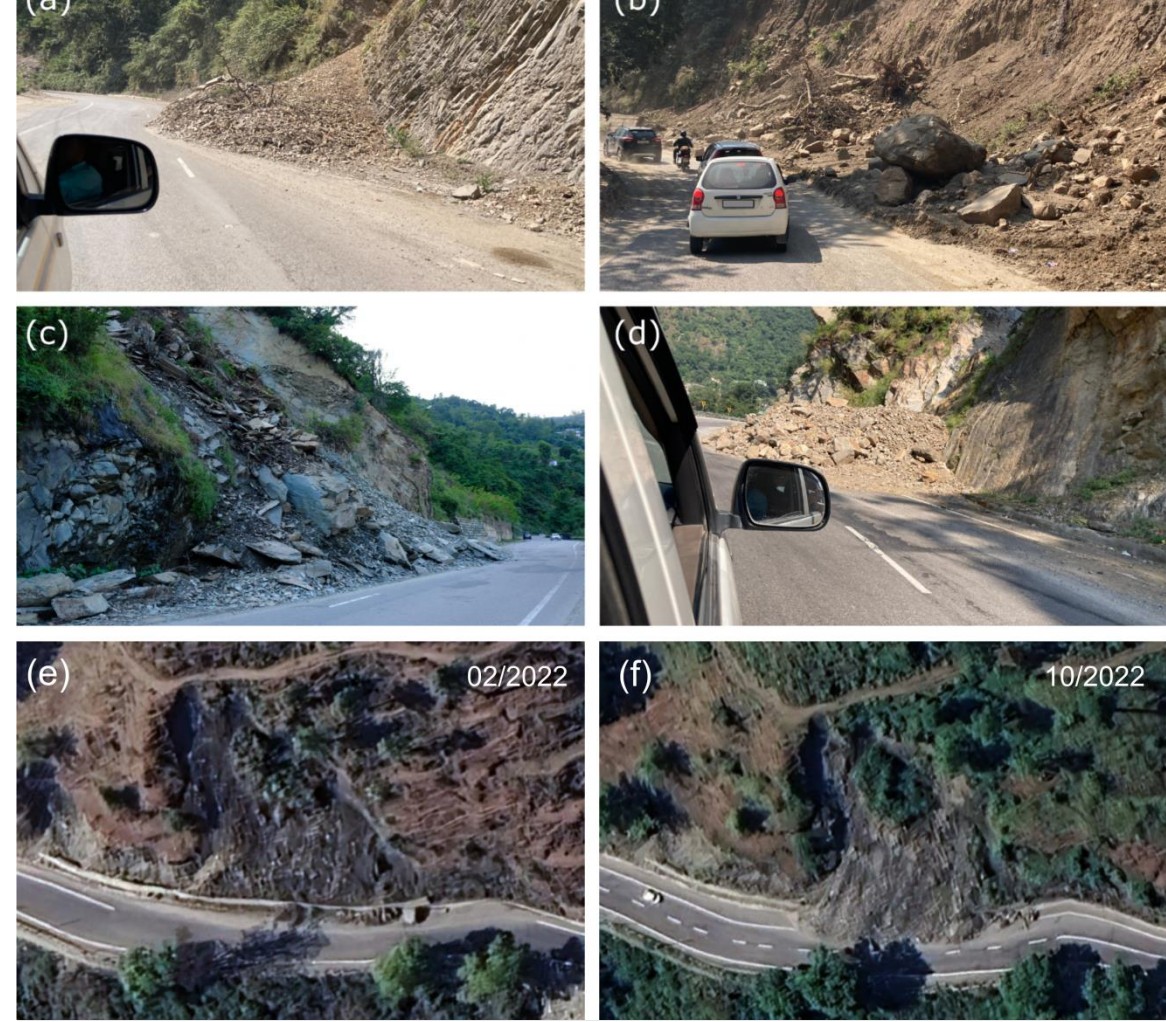

**Figure 2: Examples of partially road-blocking landslides along the highway NH-7.** Panels a) and c) show locations, where hillslopes parallel the foliation/bedding. Panels (e) and (f) show a landslide that was reactivated in the observation period (Google, ©2024 Airbus).

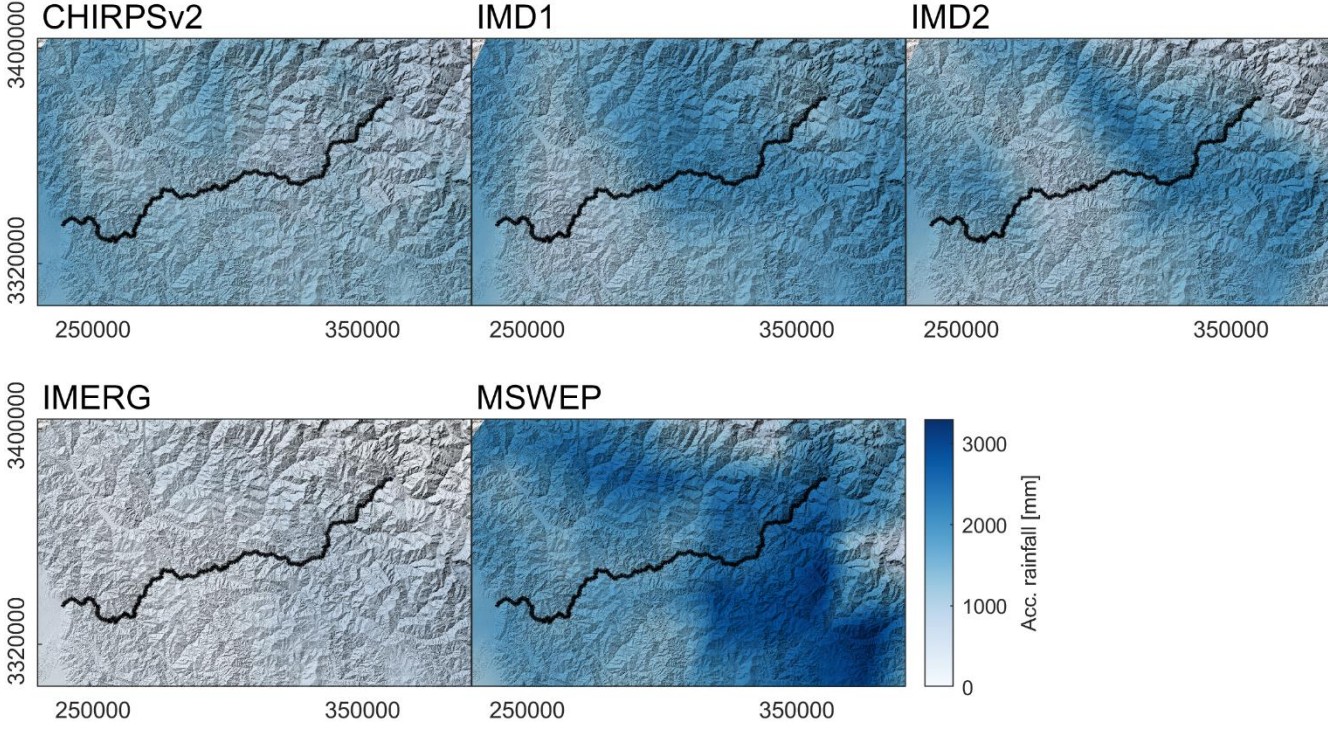

**Figure 3: Accumulated rainfall/precipitation amounts from different gridded products between Jan 1 and Oct 10, 2023.** The black line indicates the road between Rishikesh and Joshimath (see Fig. 1).

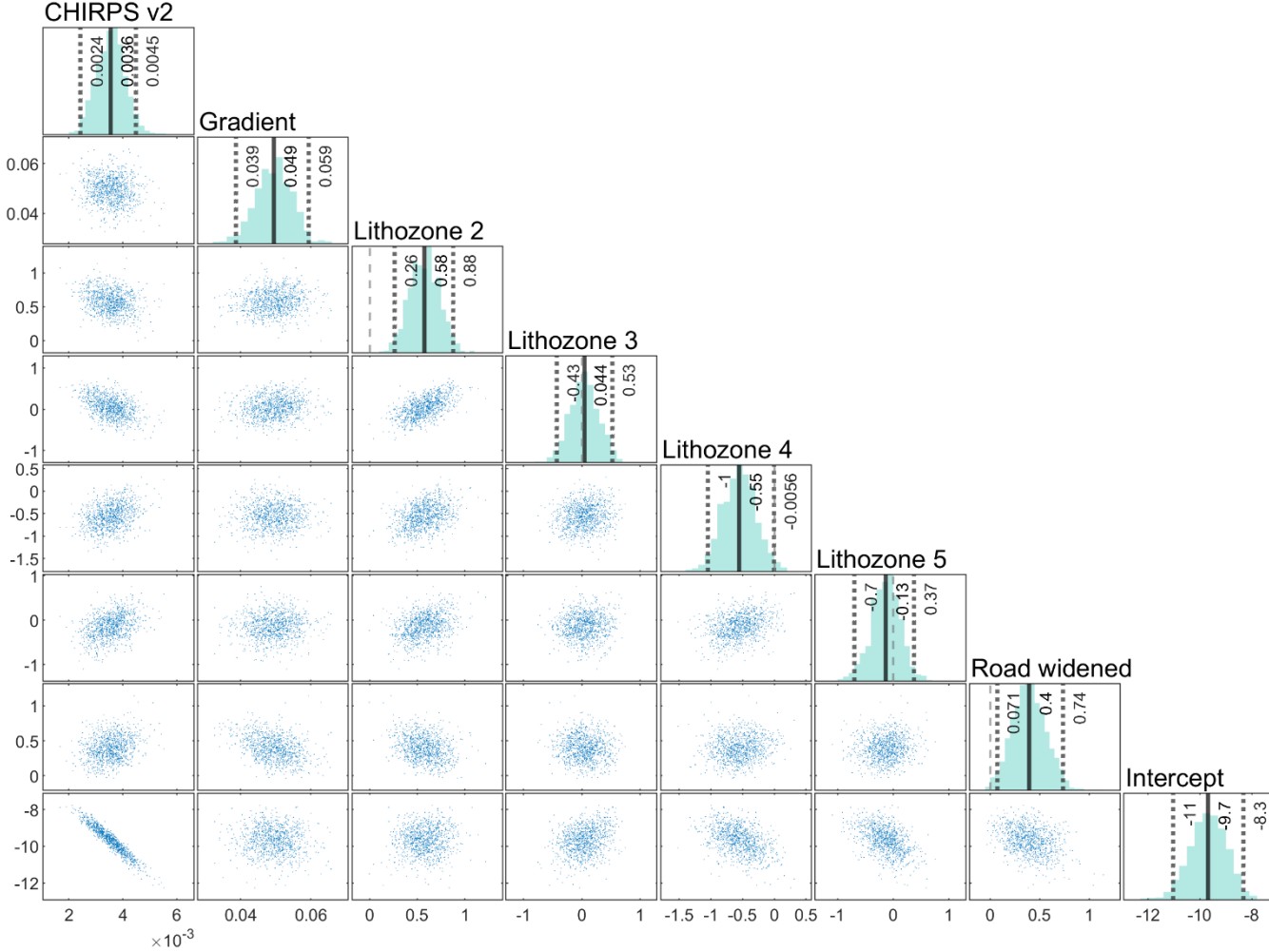

**Figure 4: Posterior parameter samples of the loglinear model of landslide occurrence along the NH-7.** CHIRPS v2 is the gridded rainfall product, gradient determines the slope within 210 m to the road and lithozones were aggregated from a geological map. Note that Lithozone 1 is missing since the parameter is encapsulated in the intercept.

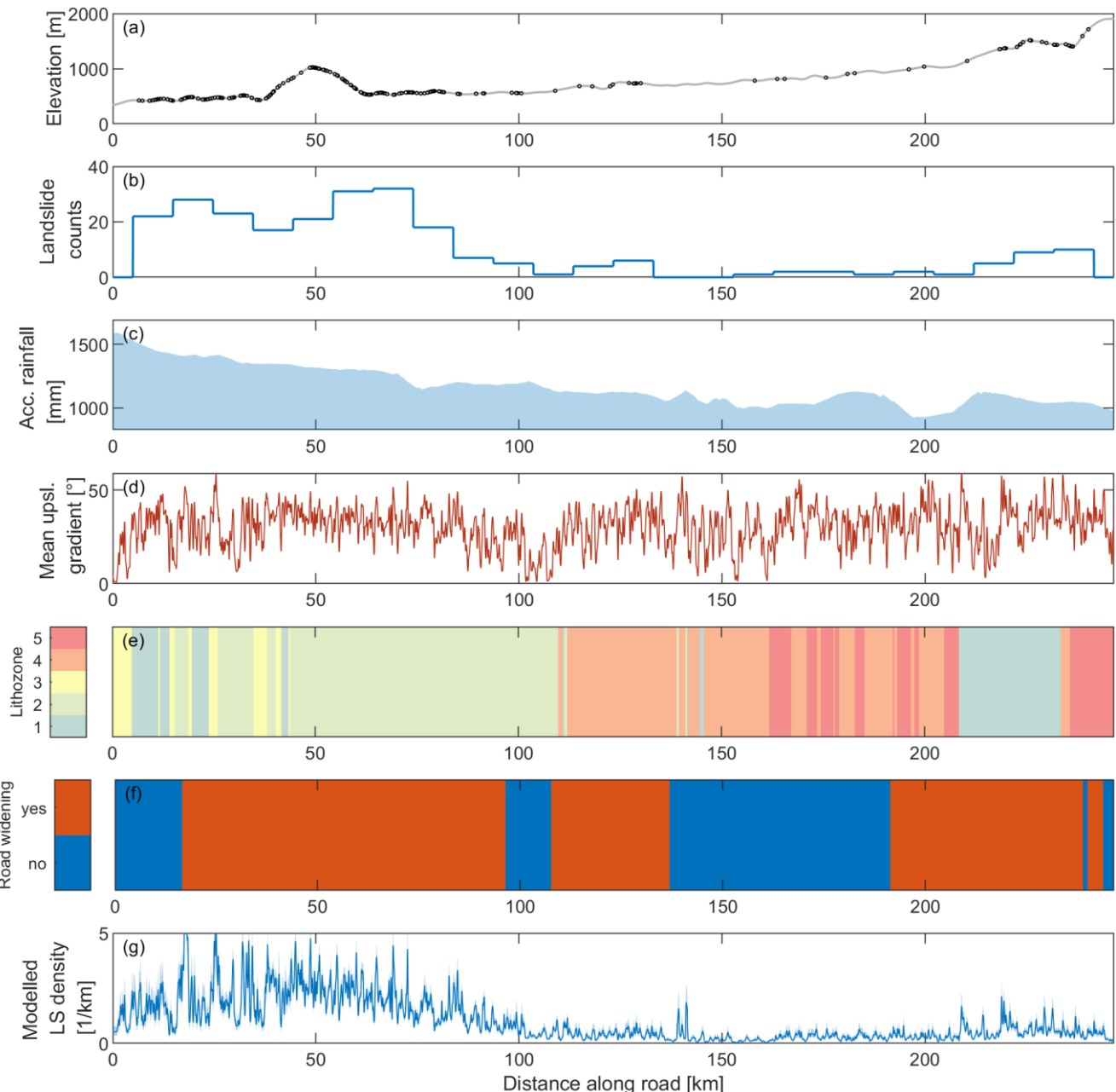

**Figure 5: Predictor and response variables used in the model.** a) shows the occurrences of the fully and partially road-blocking landslides together with the elevation profile of the road. b) Landslide density along the road measured using histogram binning (bin width is 9.8 km). c) Accumulated rainfall between January 1 and October 10, 2022 from CHIRPS v2. d) Mean upslope gradient within a distance of 210 m of the road. e) Lithozones along the road (see also Fig. 1). f) Widened road segments. g) Predicted landslide density using a model involving rainfall, slope, road widening and lithozones as covariates.

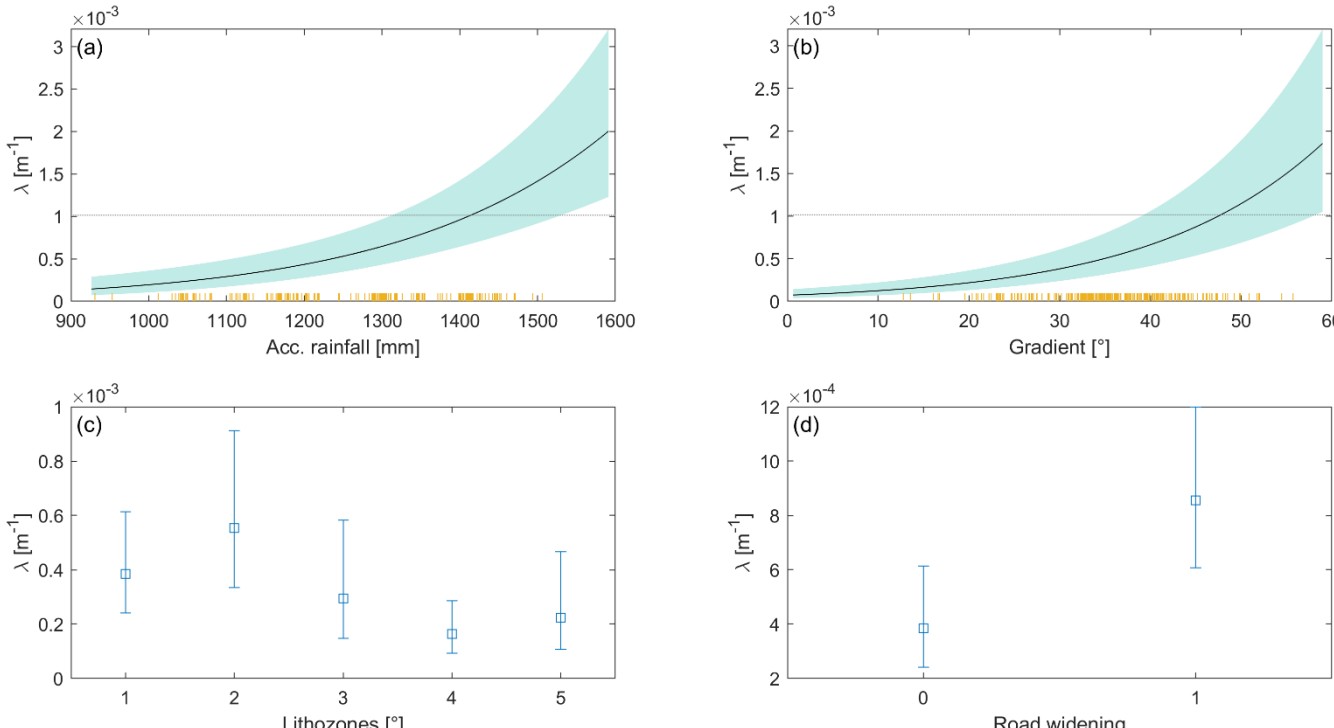

**Figure 6: Main effects of predictors in the loglinear model of road-blocking landslides along the NH-7.** a) Effect of accumulated rainfall amount between January 1 and October 10, 2022, on the occurrence of landslides, averaging out the effects of the other predictors. The orange lines indicate the occurrences of landslides. b) Effect of hillslope gradient, c) of lithozone, and d) road widening.

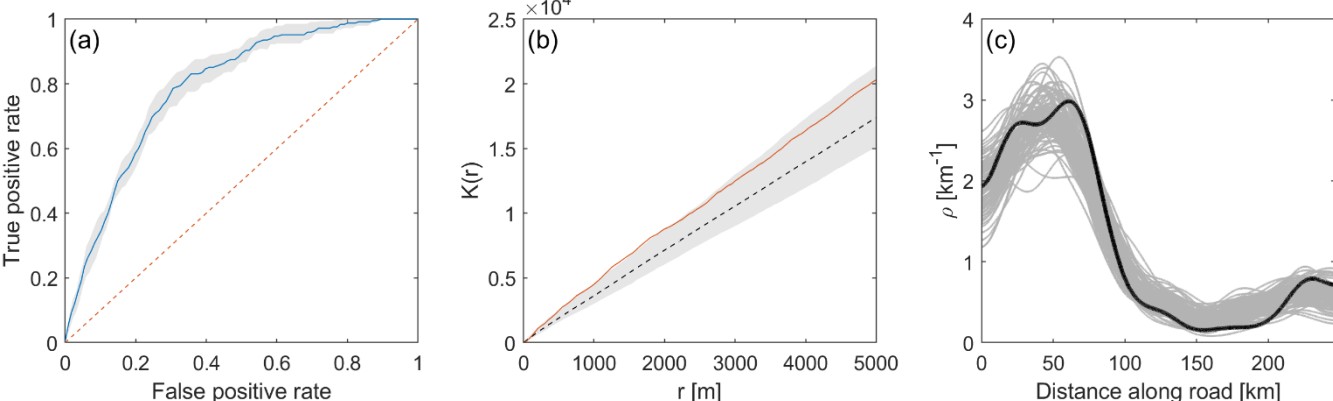

**Figure 7: Evaluation of the loglinear model including rainfall, slope, lithozones and road widening.** a) Receiver-operating-characteristics (ROC) curve. The area-under-the-curve (AUC) metric is 0.79 (0.76/0.82 95% bootstrap confidence intervals). b) The inhomogeneous K-function corrects for the influence of an inhomogeneous Poisson point process model and tests for second order effects (e.g., spatial clustering). Acceptance intervals of a theoretical model with independent point independence are shown in gray and were

705 computed from 100 bootstrap samples. The red line is the empirical inhomogeneous K-function. As the curve remains within the bootstrapped acceptance intervals, we retain the null hypothesis of inter-point independence. c) Comparison of observed landslide densities (black line) with densities obtained from 100 random realizations (gray lines) from the model.

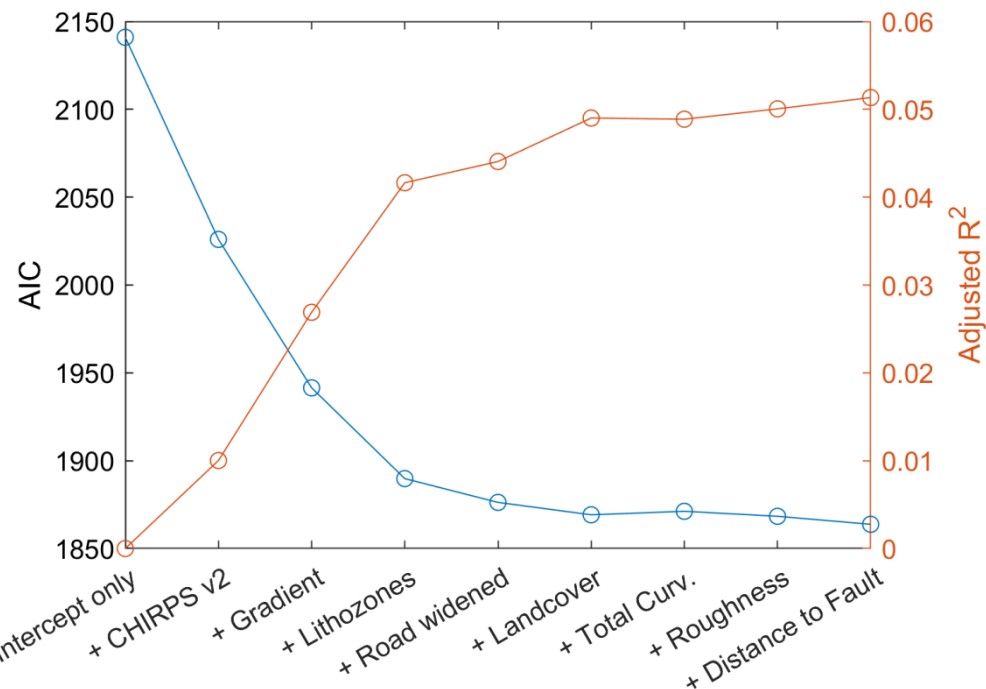

**Figure 8: Forward stepwise selection of additional explanatory covariates in the loglinear model of road-blocking landslides.**

**Tables**

**Table 1. Overview of the rainfall products.**

| Product | Product name | Spatial resolution | Link to source | Reference |
|---|---|---|---|---|
| IMD1 | Indian Meteorological Department 1 | 0.25° x 0.25° | https://doi.org/10.54302/mausam.v65i1.851 | Pai et al., 2014 |
| IMD2 | Indian Meteorological Department 2 | 0.25° x 0.25° | https://doi.org/10.2151/jmsj.87A.265 | Mitra et al., 2009 |
| MSWEP v2 | Multi-Source Weighted-Ensemble Precipitation | 0.1° x 0.1° | https://doi.org/10.5194/hess-21-6201-2017 | Beck et al., 2017 |
| CHIRPS v2 | Climate Hazards group Infrared Precipitation with Stations | 0.05° x 0.05° | https://doi.org/10.1038/sdata.2015.66 | Funk et al., 2015 |
| IMERG late run | Integrated Multi-satellitE Retrievals for GPM | 0.1° x 0.1° | https://doi.org/10.5067/GPM/IMERGDL/DAY/06 | Huffman et al., 2019 |

**Table 2. Definition of lithozones.**

| Lithozone | Aggregated lithologies | Id | Percentage of road |
|---|---|---|---|
| 1 | limestone, dolomitic limestone with shale | 2072 | 17 % |
| | shale with lenticles of limestone | 2073 | |
| | argillaceous limestone and clay | 2074 | |
| | limestone, dolomite, shale, carb. phyllite/slate | 2480 | |
| | limestone | 2457 | |
| | dolomite | 2456 | |
| 2 | shale, quartzite, limestone and conglomerate | 2109 | 33 % |
| | phyllite, qtz, shale, dolomite, tuff with dolerite | 2108 | |
| | splintery shale with nodular limestone | 746 | |
| | massive sandy limestone | 1799 | |
| | limestone, dolomite, shale and cherty quartzite | 2482 | |
| | quartzite, slate, lensoidal limestone and tuff | 2486 | |
| 3 | quartzite, limestone and occasional conglomerate | 1943 | 6 % |
| | quartzite, siltstone, chert and phosphatic shale | 1944 | |
| | diamictite, quartzite, slate and boulder bed | 2081 | |
| | carbonaceous shale, slate, greywacke | 2078 | |
| 4 | quartzite and slate with basic metavolcanics | 2464 | 30 % |
| | basic meta-volcanics | 2458 | |
| | basic / intermediate intrusive | 2453 | |
| | porphyritic nonfoliated granite | 2452 | |
| 5 | sericite quartz schist, chlorite schist | 2463 | 14 % |
| | chlorite schist, hornblende-albite-zoisite schist | 2461 | |
| | phyllite with chloritic, graphitic & carbonaceous | 2462 | |
| | schist, augen gneiss, quartzite & amphibolite | 3702 | |
| | quartz-sericite-chlorite schist & limestone | 3701 | |
| | schist, gneiss, marble and basic intrusives | 3747 | |
| | gneiss, kyanite schist, quartzite, calc-silicate | 3752 | |
| | quartzite and quartz mica schist | 3744 | |
| | calc silicate, quartzite, schist, marble band | 3743 | |

[a]Id refers to the UID given in the original data (Geological Survey of India, 2022)

**Table 3.** Aggregation of land cover classes derived from the Copernicus Global Land Service version 3 Globe 2015-2019. Remaining map codes in the original data were not concerned along the NH-7.

| Aggregated land cover class | Map codes |
|---|---|
| Closed forest | 111-116 |
| Open forest | 121-126 |
| Shrubland | 20, 30 |
| Cropland | 40 |
| Built-up | 50 |

**Table 4.** Summary of posterior distributions of parameters fitted by the Bayesian loglinear point process model. Ranks in the model were calculated according to Makalic and Schmidt (2011).

| Parameter | Mean (coefficient) | Standard deviation (coefficient) | 95% credible interval | t-Statistics | Rank |
|---|---|---|---|---|---|
| Rainfall (CHIRPS v2) | 0.00356 | 0.00055 | 0.00251 / 0.00461 | 6.502 | 1 |
| Hillslope gradient | 0.04961 | 0.00531 | 0.03842 / 0.05925 | 9.314 | 1 |
| Lithology (2) | 0.57462 | 0.16180 | 0.26311 / 0.88471 | 3.557 | 3 |
| (3) | 0.04115 | 0.24679 | -0.42837 / 0.52715 | 0.158 | 6 |
| (4) | -0.55397 | 0.26906 | -1.06779 / -0.02107 | -2-057 | 4 |
| (5) | -0.11745 | 0.26646 | -0.69276 / 0.38223 | -0.524 | 6 |
| Road widening | 0.39155 | 0.17249 | 0.07137 / 0.73991 | 2.314 | 4 |
| Intercept | -9.66776 | 0.71529 | -11.12997 / -8.39594 | - | - |