# Peer review of "More than one landslide per road kilometer – surveying and modeling mass movements along the Rishikesh-Joshimath (NH-7) highway, Uttarakhand, India"

_EGUsphere, 2023_

## Author Comment (AC1)

India Meteorological Department
Hydromet Division, New Delhi

DISTRICT - WEEK BY WEEK DEPARTURES
Period :01-10-2022 To 07-12-2022

| S NO. | MET. SUBDIVISION/UT/ STATE/DISTRICT | Week End 5-10-2022 | Week End 12-10-2022 | Week End 19-10-2022 | Week End 26-10-2022 | Week End 2-11-2022 | Week End 9-11-2022 | Week End 16-11-2022 | Week End 23-11-2022 | Week End 30-11-2022 | Week End 7-12-2022 |
|---|---|---|---|---|---|---|---|---|---|---|---|
| | SUBDIVISION : A & N ISLAND | | | | | | | | | | |
| 1 | NICOBAR | -27 | -12 | 27 | 149 | -42 | -32 | 37 | -56 | -66 | 25 |
| 2 | NORTH & MIDDLE ANDAMAN | 42 | 0 | -18 | 51 | 24 | -70 | -34 | 156 | 132 | 374 |
| 3 | SOUTH ANDAMAN | -29 | -28 | 51 | 5 | -32 | -58 | -69 | -6 | -72 | 44 |
| | SUBDIVISION : ARUNACHAL PRADESH | | | | | | | | | | |
| 1 | ANJAW | -74 | 55 | -68 | 215 | -100 | -100 | -100 | -100 | -100 | -100 |
| 2 | CHANGLANG | -52 | 6 | 88 | 513 | -58 | -100 | -100 | -100 | -100 | -100 |
| 3 | DIBANG VALLEY | -100 | | | -100 | -100 | | | | | |
| 4 | EAST KAMENG | -55 | 363 | -83 | 123 | -100 | -100 | -100 | -100 | -100 | -100 |
| 5 | EAST SIANG | -76 | 807 | 164 | 111 | 143 | -100 | -100 | -100 | -100 | -100 |
| 6 | KURUNG KUMEY | 71 | 49 | -15 | 470 | 344 | -100 | -100 | -89 | -100 | -100 |
| 7 | LOHIT | -93 | 28 | -57 | 51 | -60 | -100 | -100 | -100 | -100 | -100 |
| 8 | LOWER DIBANG VALLEY | -80 | 541 | 72 | 120 | 5 | -100 | -100 | -100 | -100 | -100 |
| 9 | LOWER SUBANSIRI | -13 | -8 | -98 | 136 | -95 | -100 | -88 | -93 | -100 | -100 |
| 10 | PAPUM-PARE | -27 | 597 | -45 | 271 | -98 | -100 | -100 | -100 | -100 | -100 |
| 11 | TAWANG | 40 | 88 | 55 | 277 | -79 | -100 | -78 | -99 | -100 | -100 |
| 12 | TIRAP | -100 | -100 | | 183 | -57 | -100 | -100 | -100 | -100 | -100 |
| 13 | UPPER SIANG | -62 | 394 | 42 | 181 | -19 | -100 | -59 | -100 | -100 | -100 |
| 14 | UPPER SUBANSIRI | -81 | -24 | -100 | 8 | 153 | -62 | -100 | -100 | -100 | -100 |
| 15 | WEST KAMENG | 5 | 147 | -95 | 392 | -66 | -96 | -100 | -100 | -100 | -100 |
| 16 | WEST SIANG | -68 | 211 | -71 | 234 | 347 | -100 | -86 | -100 | -100 | -100 |
| | SUBDIVISION : ASSAM & MEGHALAYA | | | | | | | | | | |
| | STATE : ASSAM | | | | | | | | | | |
| 1 | BAKSA | -23 | 225 | -37 | 430 | -100 | -100 | -100 | -100 | -100 | -100 |
| 2 | BARPETA | -77 | 146 | 60 | 286 | -35 | -100 | -100 | -100 | -100 | -100 |
| 3 | BONGAIGAON | -88 | 148 | -13 | 70 | -49 | -100 | -100 | -100 | -100 | -100 |
| 4 | CACHAR | -26 | -29 | -65 | 10 | -100 | -95 | -100 | -100 | -100 | -100 |
| 5 | CHIRANG | -64 | 78 | 45 | 184 | 1 | -100 | -100 | -100 | -100 | -100 |
| 6 | DARRANG | 61 | 68 | -52 | 148 | -100 | -100 | -100 | -100 | -100 | -100 |
| 7 | DHEMAJI | -39 | 770 | -77 | 64 | -86 | -100 | -100 | -100 | -100 | -100 |
| 8 | DHUBRI | -95 | 208 | 86 | 38 | -100 | -100 | -100 | -100 | -100 | -100 |
| 9 | DIBRUGARH | -11 | 208 | 53 | 84 | -94 | -100 | -100 | -100 | -100 | -100 |
| 10 | GOALPARA | -83 | 167 | 37 | 264 | -100 | -100 | -100 | -100 | -100 | -100 |
| 11 | GOLAGHAT | 52 | 65 | -18 | 158 | -98 | -100 | -100 | -100 | -100 | -100 |
| 12 | HAILAKANDI | 39 | -46 | -7 | 45 | -100 | -100 | -100 | -100 | -77 | -100 |
| 13 | JORHAT | 99 | 292 | -31 | 263 | -100 | -100 | -100 | -100 | -100 | -100 |
| 14 | KAMRUP METRO | 52 | -10 | -10 | 234 | -87 | -100 | -100 | -100 | -100 | -100 |
| 15 | KAMRUP RURAL | 12 | -48 | -22 | 409 | -100 | -100 | -100 | -100 | -100 | -100 |
| 16 | KARBI ANALOG | 67 | 37 | 6 | 150 | -100 | -100 | -100 | -100 | -100 | -100 |
| 17 | KARIMGANJ | -69 | 106 | -44 | 114 | -100 | -100 | -100 | -100 | -100 | -100 |
| 18 | KOKRAJHAR | -77 | 416 | -2 | 195 | -97 | -100 | -100 | -100 | -100 | -100 |
| 19 | LAKHIMPUR | -17 | 851 | -33 | 172 | -44 | -100 | -100 | -100 | -100 | -100 |
| 20 | MORIGAON | 25 | 53 | -96 | 200 | -100 | -72 | -100 | -100 | -100 | -100 |

| | | | | | | | | | | |
|---|---|---|---|---|---|---|---|---|---|---|
| 21 | N.C HILLS | -54 | -70 | -77 | 163 | -100 | -100 | -100 | -100 | -100 | -100 |
| 22 | NAGAON | 5 | 6 | -70 | 138 | -90 | -100 | -100 | -100 | -100 | -100 |
| 23 | NALBARI | -47 | 48 | -10 | 462 | -86 | -100 | -100 | -100 | -100 | -100 |
| 24 | SIBSAGAR | 45 | 32 | -9 | 62 | -100 | -100 | -100 | -100 | -100 | -100 |
| 25 | SONITPUR | 81 | 45 | -67 | 196 | -100 | -100 | -100 | -100 | -100 | -100 |
| 26 | TINSUKIA | -64 | 73 | -48 | 100 | -71 | -99 | -100 | -100 | -100 | -100 |
| 27 | UDALGURI | 34 | 25 | -67 | 346 | -93 | -100 | -100 | -100 | -100 | -100 |
| | STATE : MEGHALAYA | | | | | | | | | | |
| 1 | EAST GARO HILLS | -47 | 319 | 46 | 452 | -100 | -100 | -100 | -100 | -100 | -100 |
| 2 | EAST JAINTIA HILLS | -19 | -46 | -87 | 274 | -100 | -100 | -100 | -100 | -100 | -100 |
| 3 | EAST KHASI HILLS | -22 | 3 | -22 | 309 | -99 | -96 | -100 | -100 | -100 | -100 |
| 4 | NORTH GARO HILLS | -80 | 50 | -95 | 124 | -100 | -100 | -100 | -100 | -100 | -100 |
| 5 | RI BHOI | -41 | -73 | -50 | 372 | -99 | -41 | -100 | -100 | -100 | -100 |
| 6 | SOUTH GARO HILLS | 9 | -42 | 138 | 269 | -84 | -100 | -100 | -100 | -100 | -100 |
| 7 | SOUTH WEST GARO HILLS | 116 | 71 | -100 | 7 | -100 | -100 | -100 | -100 | -100 | -100 |
| 8 | SOUTH WEST KHASI HILLS | 44 | 58 | 285 | 555 | -95 | -100 | -100 | -100 | -100 | -100 |
| 9 | WEST GARO HILLS | -47 | 226 | 60 | 151 | -100 | -100 | -100 | -100 | -100 | -100 |
| 10 | WEST JAINTIA HILLS | -89 | -89 | -92 | 319 | -78 | -88 | -100 | -100 | -100 | -100 |
| 11 | WEST KHASI HILLS | -8 | 12 | -60 | 291 | -99 | -100 | -100 | -100 | -100 | -100 |
| | SUBDIVISION : N M M T | | | | | | | | | | |
| | STATE : NAGALAND | | | | | | | | | | |
| 1 | DIMAPUR | -48 | -72 | -14 | 122 | -98 | -3 | -100 | -100 | -100 | -100 |
| 2 | KIPHIRE | -75 | 10 | -69 | 201 | -100 | -100 | -100 | -100 | -100 | -100 |
| 3 | KOHIMA | -22 | -87 | -84 | 87 | -100 | -100 | -100 | -100 | -100 | -100 |
| 4 | LONGLENG | -51 | -32 | -73 | 19 | -52 | -100 | -100 | -100 | -100 | -100 |
| 5 | MOKOKCHUNG | -17 | -42 | -59 | -3 | 27 | -100 | -100 | -100 | -100 | -100 |
| 6 | MON | | | | | | | | | | |
| 7 | PEREN | -84 | -91 | 87 | 62 | -100 | -100 | -100 | -100 | -100 | -100 |
| 8 | PHEK | -19 | -30 | 4 | 274 | -100 | -100 | -100 | -100 | -100 | -100 |
| 9 | TUENSANG | -5 | -11 | -39 | -38 | -100 | -100 | -100 | -100 | -100 | -100 |
| 10 | WOKHA | -36 | -38 | 79 | 59 | -83 | -100 | -100 | -100 | -100 | |
| 11 | ZUNHEBOTO | 103 | 49 | 41 | 701 | -100 | -100 | -100 | -100 | -100 | -100 |
| | STATE : MANIPUR | | | | | | | | | | |
| 1 | BISHNUPUR | 112 | -100 | -86 | 266 | -100 | -100 | -100 | -100 | -71 | -100 |
| 2 | CHANDEL | 155 | -78 | 18 | 229 | -100 | -100 | -100 | -100 | -70 | -100 |
| 3 | CHURACHANDPUR | 65 | -92 | -65 | 152 | -100 | -100 | -100 | -100 | -50 | 200 |
| 4 | IMPHAL EAST | 42 | -100 | -99 | 180 | -100 | -97 | -100 | -100 | -100 | -100 |
| 5 | IMPHAL WEST | 22 | -100 | -6 | 226 | -99 | -100 | -100 | -100 | -100 | -100 |
| 6 | SENAPATI | -60 | -95 | -1 | 287 | -97 | -100 | -100 | -100 | -100 | 9900 |
| 7 | TAMENGLONG | 2 | 27 | -88 | 51 | -99 | -100 | -100 | -100 | -100 | -100 |
| 8 | THOUBAL | 12 | -84 | -99 | 143 | -90 | -100 | -100 | -100 | -100 | 19 |
| 9 | UKHRUL | 154 | -88 | 35 | 321 | -100 | -100 | -100 | -100 | -100 | -100 |
| | STATE : MIZORAM | | | | | | | | | | |
| 1 | AIZAWL | 10 | -84 | -33 | -19 | -100 | -100 | -100 | -100 | -85 | -100 |
| 2 | CHAMPHAI | 18 | -86 | -52 | -23 | -100 | -100 | -100 | -100 | -100 | 360 |
| 3 | KOLASIB | -16 | -65 | 30 | 48 | -100 | -92 | -100 | -100 | -59 | 10000 |
| 4 | LAWNGTLAI | -16 | 4 | -2 | -46 | -100 | -95 | -100 | -100 | -100 | 3400 |
| 5 | LUNGLEI | -61 | | -3 | 52 | -100 | -100 | -100 | -100 | -100 | 850 |
| 6 | MAMIT | -50 | -90 | -9 | 153 | -100 | -88 | -100 | -100 | -100 | -100 |
| 7 | SAIHA | 4 | 35 | 51 | 16 | -100 | -100 | -100 | -100 | -100 | -100 |
| 8 | SERCHHIP | -63 | -53 | -19 | 173 | -100 | -97 | -100 | -100 | -81 | -100 |
| | STATE : TRIPURA | | | | | | | | | | |
| 1 | DHALAI | 293 | -64 | -2 | 133 | -100 | -100 | -100 | -100 | -100 | -100 |
| 2 | GOMATI | 171 | -38 | -34 | 189 | -100 | -100 | -100 | -100 | -100 | -100 |
| 3 | KHOWAI | -17 | 75 | -77 | -45 | -100 | -100 | -100 | -100 | -100 | -100 |
| 4 | NORTH TRIPURA | 181 | -22 | -39 | 99 | -100 | -100 | -100 | -100 | -100 | -100 |

| | | | | | | | | | | |
|---|---|---|---|---|---|---|---|---|---|---|
| 5 | SIPAHIJALA | 100 | -70 | 0 | 348 | -100 | -100 | -100 | -100 | -100 | -100 |
| 6 | SOUTH TRIPURA | 128 | -99 | -93 | 92 | -100 | -100 | -100 | -100 | -100 | -100 |
| 7 | UNAKOTI | 137 | -37 | 12 | 102 | -100 | -100 | -100 | -100 | -100 | -100 |
| 8 | WEST TRIPURA | 105 | -74 | -45 | 241 | -100 | -94 | -100 | -100 | -100 | -100 |
| | SUBDIVISION : SHWB & SIKKIM | | | | | | | | | | |
| | STATE : SIKKIM | | | | | | | | | | |
| 1 | EAST SIKKIM | 61 | 108 | -79 | -100 | -100 | -100 | -71 | -100 | -100 | -100 |
| 2 | NORTH SIKKIM | 50 | 560 | -76 | -86 | -49 | -100 | -27 | -100 | -100 | -100 |
| 3 | SOUTH SIKKIM | 27 | 95 | 11 | -100 | -100 | -100 | -100 | -100 | -100 | -100 |
| 4 | WEST SIKKIM | 61 | 36 | -64 | -100 | -97 | -100 | -84 | -100 | -100 | -100 |
| | STATE : WEST BENGAL | | | | | | | | | | |
| 1 | DAKSHIN DINAJPUR | -84 | -68 | 410 | -91 | -100 | -100 | -100 | -100 | -100 | -100 |
| 2 | DARJEELING | 69 | 103 | -76 | -96 | -97 | -100 | -53 | -100 | -100 | -100 |
| 3 | JALPAIGURI | -48 | 250 | -87 | -52 | -71 | -100 | -46 | -100 | -100 | -100 |
| 4 | KOCHBIHAR | -78 | 119 | -38 | -64 | -100 | -100 | -100 | -100 | -100 | -100 |
| 5 | MALDA | 81 | -74 | 17 | -100 | -100 | -100 | -100 | -100 | -100 | -100 |
| 6 | UTTAR_DINAJPUR | 171 | -49 | -100 | -100 | -100 | -100 | -100 | -100 | -100 | -100 |
| | SUBDIVISION : GANGETIC WEST BENGAL | | | | | | | | | | |
| 1 | BANKURA | 64 | 29 | -50 | -97 | -100 | -100 | -100 | -100 | -100 | -100 |
| 2 | BARDAMAN | -10 | 26 | 36 | -99 | -100 | -100 | -100 | -100 | -100 | -100 |
| 3 | BIRBHUM | 26 | 83 | -29 | -99 | -100 | -100 | -100 | -100 | -100 | -100 |
| 4 | EASTMEDNIPUR | 82 | -28 | -90 | -84 | -100 | -100 | -100 | -100 | -100 | -100 |
| 5 | HAORA | 54 | 10 | 211 | -83 | -100 | -100 | -100 | -100 | -100 | -100 |
| 6 | HUGLI | 313 | -51 | -70 | -95 | -100 | -100 | -100 | -100 | -100 | -100 |
| 7 | KOLKATA | 28 | -10 | -34 | -81 | -100 | -100 | -100 | -100 | -100 | -100 |
| 8 | MURSIDABAD | -1 | 67 | -86 | -100 | -100 | -100 | -100 | -100 | -100 | -100 |
| 9 | NADIA | -32 | 35 | -76 | -92 | -100 | -100 | -100 | -100 | -100 | -100 |
| 10 | NORTH 24 PRAGANA | -37 | -5 | -9 | -34 | -100 | -100 | -100 | -100 | -100 | -100 |
| 11 | PASHCHIM MEDINIPUR | 158 | -66 | -28 | -99 | -100 | -100 | -100 | -100 | -100 | -100 |
| 12 | PURULIYA | 71 | 5 | 27 | -87 | -100 | -100 | -100 | -100 | -100 | -100 |
| 13 | SOUTH 24 PARGANAS | 9 | -38 | -43 | -57 | -100 | -100 | -100 | -100 | -100 | -100 |
| | SUBDIVISION : ODISHA | | | | | | | | | | |
| 1 | ANUGUL | 66 | 22 | 61 | -96 | -100 | -100 | -100 | -100 | -100 | -100 |
| 2 | BALANGIR | 25 | 98 | 15 | -100 | -100 | -100 | -100 | -100 | -100 | -100 |
| 3 | BALESHWAR | 128 | 17 | -75 | -99 | -99 | -100 | -100 | -100 | -100 | -100 |
| 4 | BARAGARH | 133 | 93 | 0 | -100 | -100 | -100 | -100 | -100 | -100 | -100 |
| 5 | BAUDA | 162 | -20 | 111 | -100 | -100 | -100 | -100 | -100 | -100 | -100 |
| 6 | BHADRAK | 165 | 1 | -4 | -99 | -100 | -100 | -100 | -100 | -100 | -100 |
| 7 | CUTTACK | 73 | 119 | -22 | -100 | -100 | -100 | -100 | -100 | -100 | -100 |
| 8 | DEOGARH | 164 | -57 | 13 | -100 | -100 | -100 | -100 | -100 | -100 | -100 |
| 9 | DHENKANAL | -10 | 27 | 18 | -75 | -100 | -100 | -100 | -100 | -100 | -100 |
| 10 | GAJAPATHI | 38 | 78 | -5 | -96 | -100 | -71 | -100 | -100 | -100 | -100 |
| 11 | GANJAM | 109 | 128 | -34 | -99 | -100 | -98 | -100 | -100 | -100 | -100 |
| 12 | JAGATSINGHAPUR | 116 | 13 | -18 | -98 | -100 | -100 | -100 | -100 | -100 | -100 |
| 13 | JAJAPUR | 54 | -18 | 18 | -100 | -100 | -100 | -100 | -100 | -100 | -100 |
| 14 | JHARSUGUDA | 77 | -8 | 240 | -100 | -100 | -100 | -100 | -100 | -100 | -100 |
| 15 | KALAHANDI | 11 | 62 | 53 | -100 | -100 | -100 | -100 | -100 | -100 | -100 |
| 16 | KANDHAMAL | 49 | 37 | 56 | -100 | -50 | -98 | -100 | -100 | -100 | -100 |
| 17 | KENDRAPARHA | 160 | 6 | -26 | -100 | -100 | -100 | -100 | -100 | -100 | -100 |
| 18 | KENDUJHAR | 28 | -36 | 39 | -100 | -100 | -100 | -100 | -100 | -100 | -100 |
| 19 | KHORDHA | -4 | 91 | -77 | -100 | -100 | -100 | -100 | -100 | -100 | -100 |
| 20 | KORAPUT | 53 | 142 | 153 | -90 | -93 | 45 | -100 | -100 | -100 | -100 |
| 21 | MALKANGIRI | 52 | -74 | -27 | -94 | -100 | -100 | -100 | -100 | -100 | -100 |
| 22 | MAYURBHANJ | 89 | 2 | -41 | -97 | -100 | -100 | -100 | -100 | -100 | -100 |
| 23 | NABARANGAPUR | -19 | 30 | 2 | -100 | -100 | -100 | -100 | -100 | -100 | -100 |

| 24 | NAYAGARH | 59 | 83 | 12 | -100 | -100 | -100 | -100 | -100 | -100 | -100 |
|----|----------|-----|-----|-----|------|------|------|------|------|------|------|
| 25 | NUAPARHA | 22 | 16 | -15 | -100 | -100 | -100 | -100 | -100 | -100 | -100 |
| 26 | PURI | 155 | 36 | -77 | -100 | -100 | -100 | -100 | -100 | -100 | -100 |
| 27 | RAYAGARHA | 59 | 4 | -3 | -99 | -100 | -21 | -100 | -100 | -100 | -100 |
| 28 | SAMBALPUR | 195 | 33 | 65 | -100 | -100 | -100 | -100 | -100 | -100 | -100 |
| 29 | SUBARNAPUR | 80 | -36 | 23 | -100 | -100 | -100 | -100 | -100 | -100 | -100 |
| 30 | SUNDARGARH | 148 | 7 | 132 | -99 | -100 | -100 | -100 | -100 | -100 | -100 |
| | SUBDIVISION : JHARKHAND | | | | | | | | | | |
| 1 | BOKARO | 149 | 7 | -58 | -94 | -100 | -100 | -100 | -100 | -100 | -100 |
| 2 | CHATRA | 108 | 25 | -23 | -100 | -100 | -100 | -100 | -100 | -100 | -100 |
| 3 | DEOGHAR | 16 | 65 | -40 | -97 | -100 | -100 | -100 | -100 | -100 | -100 |
| 4 | DHANBAD | 18 | 23 | -28 | -89 | -100 | -100 | -100 | -100 | -100 | -100 |
| 5 | DUMKA | 79 | 48 | -12 | -100 | -100 | -100 | -100 | -100 | -100 | -100 |
| 6 | EAST SINGHBHUM | 77 | 208 | 132 | -86 | -100 | -100 | -100 | -100 | -100 | -100 |
| 7 | GARHWA | 447 | 74 | -19 | -100 | -100 | -100 | -100 | -100 | -100 | -100 |
| 8 | GIRIDIH | 220 | 10 | -67 | -90 | -100 | -100 | -100 | -100 | -100 | -100 |
| 9 | GODDA | -27 | 162 | -98 | -100 | -100 | -100 | -100 | -100 | -100 | -100 |
| 10 | GUMLA | 62 | -64 | 196 | -99 | -100 | -100 | -100 | -100 | -100 | -100 |
| 11 | HAZARIBAGH | 197 | 33 | 0 | -99 | -100 | -100 | -100 | -100 | -100 | -100 |
| 12 | JAMTARA | -17 | -25 | -73 | -100 | -100 | -100 | -100 | -100 | -100 | -100 |
| 13 | KHUNTI | -34 | -52 | 85 | -100 | -100 | -100 | -100 | -100 | -100 | -100 |
| 14 | KODERMA | 184 | 110 | -54 | -36 | -100 | -100 | -100 | -100 | -100 | -100 |
| 15 | LATEHAR | 39 | 59 | 18 | -100 | -100 | -100 | -100 | -100 | -100 | -100 |
| 16 | LOHARDAGA | 47 | -47 | 244 | -100 | -100 | -100 | -100 | -100 | -100 | -100 |
| 17 | PAKUR | -4 | -32 | -81 | -100 | -100 | -100 | -100 | -100 | -100 | -100 |
| 18 | PALAMU | 178 | -4 | -24 | -100 | -100 | -100 | -100 | -100 | -100 | -100 |
| 19 | RAMGARH | 138 | -40 | 19 | -100 | -100 | -100 | -100 | -100 | -100 | -100 |
| 20 | RANCHI | 207 | 4 | 28 | -100 | -100 | -100 | -100 | -100 | -100 | -100 |
| 21 | SAHEBGANJ | 85 | -50 | -77 | -100 | -100 | -100 | -100 | -100 | -100 | -100 |
| 22 | SARAIKELA | 99 | 100 | 111 | -96 | -100 | -100 | -100 | -100 | -100 | -100 |
| 23 | SIMDEGA | 11 | 7 | 260 | -100 | -100 | -100 | -100 | -100 | -100 | -100 |
| 24 | WEST SINGHBHUM | -22 | 41 | 89 | -97 | -100 | -100 | -100 | -100 | -100 | -100 |
| | SUBDIVISION : BIHAR | | | | | | | | | | |
| 1 | ARARIYA | 114 | 135 | -71 | -100 | -100 | -100 | -100 | -100 | -100 | -100 |
| 2 | ARWAL | 127 | 84 | -100 | -100 | -100 | -100 | -100 | -100 | -100 | -100 |
| 3 | AURANGABAD | 191 | -1 | -95 | -100 | -100 | -100 | -100 | -100 | -100 | -100 |
| 4 | BANKA | 72 | 109 | -52 | -71 | -100 | -100 | -100 | -100 | -100 | -100 |
| 5 | BEGUSARAI | 274 | 104 | -81 | -100 | -100 | -100 | -100 | -100 | -100 | -100 |
| 6 | BHABUA | 68 | 105 | -96 | -100 | -100 | -100 | -100 | -100 | -100 | -100 |
| 7 | BHAGALPUR | 61 | 33 | -68 | -85 | -100 | -100 | -100 | -100 | -100 | -100 |
| 8 | BHOJPUR | 18 | 297 | -100 | -100 | -100 | -100 | -100 | -100 | -100 | -100 |
| 9 | BUXAR | 44 | 112 | -100 | -100 | -100 | -100 | -100 | -100 | -100 | -100 |
| 10 | DRABHANGA | 76 | 90 | -90 | -100 | -100 | -100 | -100 | -100 | -100 | -100 |
| 11 | GAYA | 56 | 64 | -84 | -100 | -100 | -100 | -100 | -100 | -100 | -100 |
| 12 | GOPALGANJ | -39 | 393 | -100 | -100 | -100 | -100 | -100 | -100 | -100 | -100 |
| 13 | JAHANABAD | 109 | 88 | -100 | -100 | -100 | -100 | -100 | -100 | -100 | -100 |
| 14 | JAMUI | 118 | 61 | -98 | -65 | -100 | -100 | -100 | -100 | -100 | -100 |
| 15 | KATIHAR | 107 | -31 | -52 | -100 | -100 | -100 | -100 | -100 | -100 | -100 |
| 16 | KHAGARIA | 174 | 180 | -91 | -89 | -100 | -100 | -100 | -100 | -100 | -100 |
| 17 | KISHANGANJ | 47 | 193 | -20 | -100 | -100 | -100 | -100 | -100 | -100 | -100 |
| 18 | LAKHISARAI | 141 | 124 | -70 | 392 | -100 | -100 | -100 | -100 | -100 | -100 |
| 19 | MADHEPURA | 96 | 15 | -54 | -100 | -100 | -100 | -100 | -100 | -100 | -100 |
| 20 | MADUBANI | -4 | 41 | -100 | -100 | -100 | -100 | -100 | -100 | -100 | -100 |
| 21 | MUNGER | 118 | 132 | -78 | -73 | -100 | -100 | -100 | -100 | -100 | -100 |
| 22 | MUZAFFARPUR | -41 | 131 | -100 | -82 | -100 | -100 | -100 | -100 | -100 | -100 |

| 23 | NALANDA | 87 | 164 | -61 | -100 | -100 | -100 | -100 | -100 | -100 | -100 |
| 24 | NAWADA | 61 | 81 | -77 | -89 | -100 | -100 | -100 | -100 | -100 | -100 |
| 25 | PACHIM CHAMPARAN | -56 | 609 | -100 | -100 | -100 | -100 | -100 | -100 | -100 | -100 |
| 26 | PATNA | 18 | 127 | -87 | -91 | -100 | -100 | -100 | -100 | -100 | -100 |
| 27 | PURBA CHAMPARAN | -60 | 309 | -100 | -79 | -100 | -100 | -100 | -100 | -100 | -100 |
| 28 | PURNIA | 38 | 69 | -70 | -95 | -100 | -100 | -100 | -100 | -100 | -100 |
| 29 | ROHTAS | 105 | 237 | 1 | -100 | -100 | -100 | -100 | -100 | -100 | -100 |
| 30 | SAHARSA | 160 | 40 | -43 | -57 | -100 | -100 | -100 | -100 | -100 | -100 |
| 31 | SAMASTIPUR | 75 | 25 | -85 | -51 | -100 | -100 | -100 | -100 | -100 | -100 |
| 32 | SARAN | 11 | 45 | -100 | -100 | -100 | -100 | -100 | -100 | -100 | -100 |
| 33 | SHEIKHPURA | 46 | 89 | -96 | -100 | -100 | -100 | -100 | -100 | -100 | -100 |
| 34 | SHEOHAR | -70 | 107 | -64 | -100 | -100 | -100 | -100 | -100 | -100 | -100 |
| 35 | SITAMARHI | -64 | -33 | -83 | -100 | -100 | -100 | -100 | -100 | -100 | -100 |
| 36 | SIWAN | 14 | 52 | -100 | -100 | -100 | -100 | -100 | -100 | -100 | -100 |
| 37 | SUPAUL | 62 | 148 | -98 | -100 | -100 | -100 | -100 | -100 | -100 | -100 |
| 38 | VAISHALI | 38 | 334 | -85 | -100 | -100 | -100 | -100 | -100 | -100 | -100 |
| | SUBDIVISION : EAST UTTAR PRADESH | | | | | | | | | | |
| | STATE : UTTAR PRADESH | | | | | | | | | | |
| 1 | AMBEDKARNAGAR | -92 | 1448 | -100 | -100 | -100 | -100 | -100 | -100 | -100 | -100 |
| 2 | AMETHI | -86 | 601 | -100 | -100 | -100 | -100 | -100 | -100 | -100 | -100 |
| 3 | AYODHYA | -94 | 1944 | -100 | -100 | -100 | -100 | -100 | -100 | -100 | -100 |
| 4 | AZAMGARH | -36 | 1063 | -96 | -100 | -100 | -100 | -100 | -100 | -100 | -100 |
| 5 | BAHRAICH | 50 | 1390 | -100 | -100 | -100 | -100 | -100 | -100 | -100 | -100 |
| 6 | BALLIA | -4 | 137 | -100 | -100 | -100 | -100 | -100 | -100 | -100 | -100 |
| 7 | BALRAMPUR | 68 | 1382 | -100 | -100 | -100 | -100 | -100 | -100 | -100 | -100 |
| 8 | BANDA | -93 | 679 | -100 | -100 | -100 | -100 | -100 | -100 | -100 | -100 |
| 9 | BARABANKI | -2 | 2345 | -100 | -100 | -100 | -100 | -100 | -100 | -100 | -100 |
| 10 | BASTI | -88 | 1605 | -100 | -100 | -100 | -100 | -100 | -100 | -100 | -100 |
| 11 | BHADOHI | 137 | 724 | -100 | -100 | -100 | -100 | -100 | -100 | -100 | -100 |
| 12 | CHANDAULI | -87 | 195 | -100 | -100 | -100 | -100 | -100 | -100 | -100 | -100 |
| 13 | CHITRAKOOT | -6 | 2466 | -100 | -100 | -100 | -100 | -100 | -100 | -100 | -100 |
| 14 | DEORIA | -5 | 1721 | -100 | -100 | -100 | -100 | -100 | -100 | -100 | -100 |
| 15 | FARRUKHABAD | -100 | 760 | -100 | -100 | -100 | -100 | -100 | -100 | -100 | -100 |
| 16 | FATEHPUR | -100 | 987 | -100 | -100 | -100 | -100 | -100 | -100 | -100 | -100 |
| 17 | GAZIPUR | -20 | -28 | -100 | -100 | -100 | -100 | -100 | -100 | -100 | -100 |
| 18 | GONDA | -52 | 1567 | -100 | -100 | -100 | -100 | -100 | -100 | -100 | -100 |
| 19 | GORAKHPUR | -88 | 1205 | -100 | -100 | -100 | -100 | -100 | -100 | -100 | -100 |
| 20 | HARDOI | -100 | 1327 | -100 | -100 | -100 | -100 | -100 | -100 | -100 | -100 |
| 21 | JAUNPUR | 12 | 369 | -27 | -100 | -100 | -100 | -100 | -100 | -100 | -100 |
| 22 | KANNAUJ | -100 | 943 | -100 | -100 | -100 | -100 | -100 | -100 | -100 | -100 |
| 23 | KANPUR | -31 | 1121 | -100 | -100 | -100 | -100 | -100 | -100 | -100 | -100 |
| 24 | KANPUR DEHAT | -100 | 641 | -100 | -100 | -100 | -100 | -100 | -100 | -100 | -100 |
| 25 | KAUSHAMBI | -75 | 763 | -100 | -100 | -100 | -100 | -100 | -100 | -100 | -100 |
| 26 | KHERI | -100 | 1686 | -100 | -100 | -100 | -100 | -100 | -100 | -100 | -100 |
| 27 | KUSHINAGAR | -51 | 609 | -97 | -100 | -100 | -100 | -100 | -100 | -100 | -100 |
| 28 | LUCKNOW | -83 | 1575 | -100 | -100 | -100 | -100 | -100 | -100 | -100 | -100 |
| 29 | MAHARAJGANJ | -79 | 1641 | -100 | -100 | -100 | -100 | -100 | -100 | -100 | -100 |
| 30 | MAU | -9 | 54 | -100 | -100 | -100 | -100 | -100 | -100 | -100 | -100 |
| 31 | MIRZAPUR | -15 | 102 | -100 | -100 | -100 | -100 | -100 | -100 | -100 | -100 |
| 32 | PRATAPGARH | 27 | 2517 | -100 | -100 | -100 | -100 | -100 | -100 | -100 | -100 |
| 33 | PRAYAGRAJ | -75 | 381 | -100 | -100 | -100 | -100 | -100 | -100 | -100 | -100 |
| 34 | RAIBEARELI | -82 | 561 | -100 | -100 | -100 | -100 | -100 | -100 | -100 | -100 |
| 35 | SANTKABIRNAGAR | -19 | 906 | -100 | -100 | -100 | -100 | -100 | -100 | -100 | -100 |
| 36 | SHRAWASTI | 94 | 1890 | -100 | -100 | -100 | -100 | -100 | -100 | -100 | -100 |
| 37 | SIDDHARTHNAGAR | -78 | 2343 | -100 | -100 | -100 | -100 | -100 | -100 | -100 | -100 |

| | | | | | | | | | | |
|---|---|---|---|---|---|---|---|---|---|---|
| 38 | SITAPUR | -100 | 909 | -100 | -100 | -100 | -100 | -100 | -100 | -100 | -100 |
| 39 | SONBHADRA | 93 | 44 | -44 | -100 | -100 | -100 | -100 | -100 | -100 | -100 |
| 40 | SULTANPUR | -93 | 1098 | -100 | -100 | -100 | -100 | -100 | -100 | -100 | -100 |
| 41 | UNNAO | -93 | 800 | -100 | -100 | -100 | -100 | -100 | -100 | -100 | -100 |
| 42 | VARANASI | 23 | 221 | -97 | -100 | -100 | -100 | -100 | -100 | -100 | -100 |
| | SUBDIVISION : WEST UTTAR PRADESH | | | | | | | | | | |
| 1 | AGRA | -99 | 1398 | -100 | -100 | -100 | 1617 | -100 | -100 | -100 | -100 |
| 2 | ALIGARH | -100 | 1711 | -100 | -100 | -100 | -100 | -100 | -100 | -100 | -100 |
| 3 | AMROHA | -100 | 1309 | -100 | -100 | -100 | -100 | -100 | -100 | -100 | -100 |
| 4 | AURAIYA | -53 | 2078 | -100 | -100 | -100 | -100 | -100 | -100 | -100 | -100 |
| 5 | BADAUN | -100 | 2422 | -100 | -100 | -100 | -100 | -100 | -100 | -100 | -100 |
| 6 | BAGHPAT | -100 | 770 | -100 | -100 | -100 | -100 | -100 | -100 | -100 | -100 |
| 7 | BAREILLY | -100 | 3661 | -100 | -100 | -100 | -100 | -100 | -100 | -100 | -100 |
| 8 | BIJNOR | -100 | 2616 | -100 | -100 | -100 | -100 | -100 | -100 | -100 | -100 |
| 9 | BULANDSAHAR | -100 | 2352 | -100 | -100 | -100 | -100 | -100 | -100 | -100 | -100 |
| 10 | ETAH | -80 | 3239 | -100 | -100 | -100 | 233 | -100 | -100 | -100 | -100 |
| 11 | ETAWAH | -50 | 1685 | -100 | -100 | -100 | -100 | -100 | -100 | -100 | -100 |
| 12 | FIROZABAD | -100 | 930 | -100 | -100 | -100 | 2556 | -100 | -100 | -100 | -100 |
| 13 | GAUTAMBUDHNAGAR | -100 | 3900 | -100 | -100 | -100 | -100 | -100 | -100 | -100 | -100 |
| 14 | GHAZIABAD | -100 | 3862 | -100 | -100 | -100 | -100 | -100 | -100 | -100 | -100 |
| 15 | HAMIRPUR | -43 | 1030 | -100 | -100 | -100 | -100 | -100 | -100 | -100 | -100 |
| 16 | HAPUR | -100 | 3140 | -100 | -100 | -100 | -100 | -100 | -100 | -100 | -100 |
| 17 | HATHRAS | -100 | 2276 | -100 | -100 | -100 | 175 | -100 | -100 | -100 | -100 |
| 18 | JALAUN | -72 | 1962 | -100 | -100 | -100 | -100 | -100 | -100 | -100 | -100 |
| 19 | JHANSI | -71 | 644 | -56 | -100 | -100 | -100 | -100 | -100 | -100 | -100 |
| 20 | KASGANJ | -100 | 1890 | -100 | -100 | -100 | -100 | -100 | -100 | -100 | -100 |
| 21 | LALITPUR | -100 | 512 | -100 | -100 | -100 | -100 | -100 | -100 | -100 | -100 |
| 22 | MAHOBA | 139 | 991 | -100 | -100 | -100 | -100 | -100 | -100 | -100 | -100 |
| 23 | MAINPURI | -88 | 1158 | -100 | -100 | -100 | -100 | 0 | -100 | -100 | -100 |
| 24 | MATHURA | -100 | 1763 | -100 | -100 | -100 | 275 | -100 | -100 | -100 | -100 |
| 25 | MEERUT | -100 | 3365 | -100 | -100 | -100 | -100 | -100 | -100 | -100 | -100 |
| 26 | MORADABAD | -100 | 2240 | -100 | -100 | -100 | -100 | -100 | -100 | -100 | -100 |
| 27 | MUZAFARNAGAR | -100 | 1952 | -100 | -100 | -100 | -100 | -100 | -100 | -100 | -100 |
| 28 | PILHIBHIT | -100 | 1196 | -100 | -100 | -100 | -100 | -100 | -100 | -100 | -100 |
| 29 | RAMPUR | -100 | 1942 | -100 | -100 | -100 | -100 | -100 | -100 | -100 | -100 |
| 30 | SAMBHAL | -100 | 1973 | -100 | -100 | -100 | -100 | -100 | -100 | -100 | -100 |
| 31 | SHAHJAHANPUR | -100 | 1280 | -100 | -100 | -100 | -100 | -100 | -100 | -100 | -100 |
| 32 | SHAMLI | -100 | 233 | -100 | -100 | -100 | -100 | -100 | -100 | -100 | -100 |
| 33 | SHARANPUR | -100 | 168 | -100 | -100 | -100 | -100 | -100 | -100 | -100 | -100 |
| | SUBDIVISION : UTTARAKHAND | | | | | | | | | | |
| 1 | ALMORA | -78 | 2499 | -64 | 138 | -100 | -100 | -100 | -100 | -100 | -100 |
| 2 | BAGESHWAR | -49 | 2619 | -59 | 282 | 820 | -67 | -100 | -100 | -100 | -100 |
| 3 | CHAMOLI | -67 | 1855 | 20 | -63 | -48 | -100 | 67 | -100 | -100 | -100 |
| 4 | CHAMPAWAT | -96 | 1253 | -85 | 70 | -100 | -100 | -64 | -100 | -100 | -100 |
| 5 | DEHRADUN | -15 | 317 | -56 | -85 | -100 | -100 | -30 | -100 | -100 | -100 |
| 6 | HARIDWAR | -100 | 763 | -100 | -100 | -100 | -100 | -100 | -100 | -100 | -100 |
| 7 | NANITAL | -17 | 2308 | -100 | -95 | -100 | -100 | -100 | -100 | -100 | -100 |
| 8 | PAURI GARHWAL | -99 | 679 | -83 | -90 | -100 | -100 | -45 | -100 | -100 | -100 |
| 9 | PITHORAGARH | -40 | 912 | -78 | -59 | -50 | -100 | -77 | -100 | -100 | -100 |
| 10 | RUDRAPRAYAG | -20 | 218 | 23 | -55 | -100 | -100 | 342 | -100 | -100 | -100 |
| 11 | TEHRI GARWAL | -74 | 419 | -94 | -95 | -100 | -100 | 250 | -100 | -100 | -100 |
| 12 | UDHAM SINGH NAGAR | -100 | 2218 | -100 | -100 | -100 | -100 | -100 | -100 | -100 | -100 |
| 13 | UTTARKASHI | -63 | 88 | 326 | -11 | -100 | -100 | 264 | -100 | -100 | -100 |
| | SUBDIVISION : HAR. CHD & DELHI | | | | | | | | | | |

| | STATE : HARYANA | | | | | | | | | |
|---|---|---|---|---|---|---|---|---|---|---|
| 1 | AMBALA | -100 | 412 | -100 | -100 | -100 | -100 | -78 | -100 | -100 | -100 |
| 2 | BHIWANI | -100 | 541 | -100 | -100 | -100 | -100 | -100 | -100 | -100 | -100 |
| 3 | CHARKHI DADRI | -100 | 1336 | -100 | -100 | -100 | 1233 | -100 | -100 | -100 | -100 |
| 4 | FARIDABAD | -100 | 2478 | -88 | -100 | -100 | -100 | -100 | -100 | -100 | -100 |
| 5 | FATEHABAD | -100 | 1622 | -100 | -100 | -100 | -100 | -100 | -100 | -100 | -100 |
| 6 | GURGAON | -100 | 2087 | -100 | -100 | -100 | -100 | -100 | -100 | -100 | -100 |
| 7 | HISAR | -100 | 313 | -97 | -100 | -100 | -51 | -100 | -100 | -100 | -100 |
| 8 | JHAJJAR | -100 | 1591 | -96 | -100 | -100 | -100 | -100 | -100 | -100 | -100 |
| 9 | JIND | -100 | 376 | -100 | -100 | -100 | -100 | -100 | -100 | -100 | -100 |
| 10 | KAITHAL | -100 | 355 | -98 | -100 | -100 | -100 | -100 | -100 | -100 | -100 |
| 11 | KARNAL | -100 | 235 | -100 | -100 | -100 | -100 | -100 | -100 | -100 | -100 |
| 12 | KURUKSHETRA | -100 | 953 | -100 | -100 | -100 | -100 | -100 | -100 | -100 | -100 |
| 13 | MAHENDRAGARH | -100 | 1938 | -100 | -100 | -100 | 84 | -100 | -100 | -100 | -100 |
| 14 | NUH | -100 | 2509 | -100 | -100 | -100 | -100 | -17 | -100 | -100 | -100 |
| 15 | PALWAL | -100 | 3900 | -100 | -100 | -100 | -100 | -100 | -100 | -100 | -100 |
| 16 | PANCHKULA | -100 | 812 | -100 | -100 | -100 | -100 | -100 | -100 | -100 | -100 |
| 17 | PANIPAT | -100 | 717 | -100 | -100 | -100 | -100 | -100 | -100 | -100 | -100 |
| 18 | REWARI | -100 | 1883 | -100 | -100 | -100 | -83 | -100 | -100 | -100 | -100 |
| 19 | ROHTAK | -100 | 575 | -100 | -100 | -100 | -100 | -100 | -100 | -100 | -100 |
| 20 | SIRSA | -100 | 43 | -100 | -100 | -100 | -94 | -100 | -100 | -100 | -100 |
| 21 | SONIPAT | -100 | 396 | -100 | -100 | -100 | -100 | -100 | -100 | -100 | -100 |
| 22 | YAMUNANAGAR | -100 | 486 | -100 | -100 | -100 | -100 | -100 | -100 | -100 | -100 |
| | STATE : CHANDIGARH (UT) | | | | | | | | | | |
| 1 | CHANDIGARH | -100 | 495 | -100 | -100 | -100 | -100 | -95 | -100 | -100 | -100 |
| | STATE : DELHI (UT) | | | | | | | | | | |
| 1 | CENTRAL DELHI | -100 | 2711 | -100 | -100 | -100 | -100 | -100 | -100 | -100 | -100 |
| 2 | EAST DELHI | -100 | 2874 | -100 | -100 | -100 | -100 | -100 | -100 | -100 | -100 |
| 3 | NEW DELHI | -100 | 1983 | -100 | -100 | -100 | -100 | -100 | -100 | -100 | -100 |
| 4 | NORTH DELHI | -100 | 2557 | -100 | -100 | -100 | -100 | -100 | -100 | -100 | -100 |
| 5 | NORTH EAST DELHI | -100 | 1436 | -100 | -100 | -100 | -100 | -100 | -100 | -100 | -100 |
| 6 | NORTH WEST DELHI | -100 | 813 | -100 | -100 | -100 | -100 | -100 | -100 | -100 | -100 |
| 7 | SOUTH DELHI | -100 | 1907 | -100 | -100 | -100 | -100 | -100 | -100 | -100 | -100 |
| 8 | SOUTH WEST DELHI | -100 | 1484 | -100 | -100 | -100 | -100 | -100 | -100 | -100 | -100 |
| 9 | WEST DELHI | -100 | 710 | -100 | -100 | -100 | -100 | -100 | -100 | -100 | -100 |
| | SUBDIVISION : PUNJAB | | | | | | | | | | |
| 1 | AMRITSAR | -100 | -73 | -100 | -100 | -100 | -100 | -68 | -100 | -100 | -100 |
| 2 | BARNALA | -100 | -100 | -100 | -100 | -100 | -100 | -100 | -100 | -100 | -100 |
| 3 | BATHINDA | -100 | -100 | -100 | -100 | -100 | -100 | 11 | -100 | -100 | -100 |
| 4 | FARIDKOT | -100 | -100 | -100 | -100 | -100 | -100 | 0 | -100 | -100 | -100 |
| 5 | FATEHGARH SAHIB | -100 | 665 | -96 | -100 | -100 | -100 | -100 | -100 | -100 | -100 |
| 6 | FAZILKA | -100 | -100 | -100 | -100 | -100 | -100 | -100 | -100 | -100 | -100 |
| 7 | FIROZPUR | -100 | -100 | -100 | -100 | -100 | -100 | 15 | -100 | -100 | -100 |
| 8 | GURDASPUR | -100 | -66 | -100 | -100 | -100 | 308 | 364 | -100 | -100 | -100 |
| 9 | HOSHIARPUR | -100 | 93 | -100 | -100 | -100 | 5 | -88 | -100 | -100 | -100 |
| 10 | JALANDHAR | -100 | 539 | -100 | -100 | -100 | -100 | -57 | -100 | -100 | -100 |
| 11 | KAPURTHALA | -100 | 385 | -100 | -100 | -100 | -100 | 173 | -100 | -100 | -100 |
| 12 | LUDHIANA | -100 | 747 | -100 | -100 | -100 | -100 | -37 | -100 | -100 | -100 |
| 13 | MANSA | -100 | 600 | -100 | -100 | -100 | -100 | -100 | -100 | -100 | -100 |
| 14 | MOGA | -100 | -17 | -100 | -100 | -100 | -100 | -100 | -100 | -100 | -100 |
| 15 | MUKTSAR | -100 | -100 | -100 | -100 | -100 | -100 | -58 | -100 | -100 | -100 |
| 16 | PATHANKOT | -100 | 569 | -100 | 744 | -100 | 156 | 445 | -100 | -100 | -100 |
| 17 | PATIALA | -100 | 604 | -97 | -100 | -100 | -100 | -60 | -100 | -100 | -100 |
| 18 | RUPNAGAR | -100 | 371 | -13 | -100 | -100 | -100 | -100 | -100 | -100 | -100 |
| 19 | SANGRUR | -100 | 604 | -96 | -100 | -100 | -100 | -100 | -100 | -100 | -100 |
| 20 | SAS NAGAR | -100 | 42 | -100 | -100 | -100 | -100 | -100 | -100 | -100 | -100 |

| # | Name | | | | | | | | | |
|---|------|------|------|------|------|------|------|------|------|------|
| 21 | SBS NAGAR | -100 | 1240 | -100 | -100 | -100 | -100 | 22 | -100 | -100 | -100 |
| 22 | TARN TARAN | -100 | -97 | -100 | -100 | -100 | -100 | -22 | -100 | -100 | -100 |
| | **SUBDIVISION : HIMACHAL PRADESH** | | | | | | | | | | |
| 1 | BILASPUR | -97 | 1461 | 98 | -100 | -100 | -100 | -61 | -100 | -100 | -100 |
| 2 | CHAMBA | -94 | 169 | -64 | 195 | -100 | -51 | 772 | -100 | -100 | -100 |
| 3 | HAMIRPUR | -100 | 117 | -100 | -62 | -100 | -100 | -31 | -100 | -100 | -100 |
| 4 | KANGRA | -89 | 410 | -99 | 116 | -100 | -99 | 227 | -100 | -100 | -100 |
| 5 | KINNAUR | -70 | 91 | -92 | -100 | -100 | -100 | 358 | -100 | -100 | -100 |
| 6 | KULLU | -66 | 819 | -40 | -98 | -100 | -100 | 393 | -100 | -100 | -100 |
| 7 | LAHUL&SPITI | -100 | -34 | -90 | 61 | -76 | -74 | 586 | -100 | -100 | -100 |
| 8 | MANDI | -20 | 654 | -41 | -58 | -100 | -100 | 92 | -100 | -100 | -100 |
| 9 | SHIMLA | 82 | 513 | -49 | -77 | -100 | -100 | 159 | -100 | -100 | -100 |
| 10 | SIRMAUR | -81 | 628 | 12 | -100 | -100 | -100 | -93 | -100 | -100 | -100 |
| 11 | SOLAN | -65 | 453 | -41 | -100 | -100 | -100 | -69 | -100 | -100 | -100 |
| 12 | UNA | -100 | 90 | -100 | -100 | -100 | -98 | -100 | -100 | -100 | -100 |
| | **SUBDIVISION : JAMMU & KASHMIR AND LADAKH** | | | | | | | | | | |
| | **STATE : JAMMU & KASHMIR (UT)** | | | | | | | | | | |
| 1 | ANANTNAG | 32 | -79 | -59 | 137 | -60 | 194 | 79 | -99 | -100 | -100 |
| 2 | BADGAM | -85 | -100 | -85 | 88 | -69 | 268 | 87 | -100 | -100 | -100 |
| 3 | BANDIPORE | -43 | -100 | -100 | 31 | -89 | 186 | 200 | -80 | -100 | -100 |
| 4 | BARAMULA | -69 | -100 | -84 | 105 | -70 | 447 | 485 | -97 | -100 | -100 |
| 5 | DODA | -86 | -34 | -19 | 398 | -10 | 16 | 769 | -100 | -100 | -100 |
| 6 | GANDERWAL | -65 | -90 | -93 | 403 | -79 | 142 | 89 | -84 | -100 | -100 |
| 7 | JAMMU | -100 | 41 | -100 | -12 | -98 | 429 | 55 | -100 | -100 | -100 |
| 8 | KATHUA | -100 | 13 | -100 | 876 | -100 | 160 | 23 | -100 | -100 | -100 |
| 9 | KISTWAR | -77 | 158 | -58 | 445 | -71 | 107 | 733 | -100 | -100 | -100 |
| 10 | KULGAM | 0 | -86 | -87 | 414 | -71 | 361 | -40 | -100 | -100 | -100 |
| 11 | KUPWARA | -94 | -100 | -99 | 16 | -99 | 147 | 172 | -92 | -100 | -100 |
| 12 | PULWAMA | -25 | -100 | -52 | 124 | -29 | 261 | -39 | -100 | -100 | -100 |
| 13 | PUNCH | -100 | -100 | -100 | -100 | -100 | 1270 | 270 | -30 | -100 | -100 |
| 14 | RAJOURI | 176 | 491 | -49 | 197 | -100 | 38 | 686 | -100 | -100 | -100 |
| 15 | RAMBAN | -77 | 93 | -91 | 200 | -96 | 175 | 337 | -100 | -100 | -100 |
| 16 | RIASI | -100 | 474 | -100 | 146 | -100 | 528 | 487 | -100 | -100 | -100 |
| 17 | SAMBA | -100 | -100 | -100 | -76 | -100 | 61 | 4650 | -100 | -100 | -100 |
| 18 | SHOPIAN | -72 | -100 | -86 | -55 | -91 | 71 | -12 | -100 | -100 | -100 |
| 19 | SRINAGAR | -88 | -96 | -89 | 183 | -32 | 456 | 136 | -100 | -100 | -100 |
| 20 | UDHAMPUR | -95 | 195 | -95 | 71 | -100 | 427 | 68 | -100 | -100 | -100 |
| | **STATE : LADAKH (UT)** | | | | | | | | | | |
| 1 | KARGIL | -100 | -100 | -100 | -100 | 7900 | -100 | 1050 | -100 | -100 | -100 |
| 2 | LEH AND LADAKH | -100 | -79 | 81 | -33 | -33 | -97 | 1667 | -42 | -100 | -100 |
| | **SUBDIVISION : WEST RAJASTHAN** | | | | | | | | | | |
| | **STATE : RAJASTHAN** | | | | | | | | | | |
| 1 | BARMER | -100 | -100 | -100 | -100 | -100 | -100 | -100 | -100 | -100 | -100 |
| 2 | BIKANER | -100 | -100 | -100 | -100 | -100 | -75 | -50 | -100 | -100 | -100 |
| 3 | CHURU | -93 | 405 | -100 | -100 | -100 | 31 | 71 | -100 | -100 | -100 |
| 4 | GANGANAGAR | -100 | -100 | -100 | -100 | -100 | -20 | 67 | -100 | -100 | -100 |
| 5 | HANUMANGARH | -100 | -100 | -100 | -100 | -100 | -78 | -100 | -100 | -100 | -100 |
| 6 | JAISELMER | -100 | -100 | -100 | -100 | -100 | -100 | -100 | -100 | -100 | -100 |
| 7 | JALOR | -100 | -83 | -100 | -100 | -100 | -100 | -100 | -100 | -100 | -100 |
| 8 | JODHPUR | -63 | 62 | -100 | -100 | -100 | -100 | -100 | -100 | -100 | -100 |
| 9 | NAGAUR | -29 | 826 | -100 | -100 | -100 | 80 | -25 | -100 | -100 | -100 |
| 10 | PALI | 31 | 235 | -100 | -100 | -100 | -100 | -100 | -100 | -100 | -100 |
| | **SUBDIVISION : EAST RAJASTHAN** | | | | | | | | | | |

| | | | | | | | | | | |
|---|---|---|---|---|---|---|---|---|---|---|
| 1 | AJMER | -92 | 415 | -100 | -100 | -100 | -100 | -100 | -100 | -100 | -100 |
| 2 | ALWAR | -100 | 1768 | -100 | -100 | -100 | 330 | -49 | -100 | -100 | -100 |
| 3 | BANSWARA | -92 | 311 | -100 | -100 | -100 | -100 | -100 | -100 | -100 | -100 |
| 4 | BARAN | -97 | 908 | -100 | -100 | -100 | -100 | -100 | -100 | -100 | -100 |
| 5 | BHARATPUR | -98 | 2311 | -100 | -100 | -100 | 1150 | -75 | -100 | -100 | -100 |
| 6 | BHILWARA | -61 | 635 | -100 | -100 | -100 | -92 | -100 | -100 | -100 | -100 |
| 7 | BUNDI | -100 | 1067 | -100 | -100 | -100 | -100 | -100 | -100 | -100 | -100 |
| 8 | CHITTAURGARH | 10 | 972 | -100 | -100 | -100 | -100 | -100 | -100 | -100 | -100 |
| 9 | DAUSA | -100 | 2275 | -100 | -100 | -100 | 600 | -100 | -100 | -100 | -100 |
| 10 | DHOLPUR | -100 | 1513 | -100 | -100 | -100 | 11 | -100 | -100 | -100 | -100 |
| 11 | DUNGARPUR | -98 | 256 | -100 | -100 | -100 | -100 | -100 | -100 | -100 | -100 |
| 12 | JAIPUR | -98 | 1491 | -100 | -100 | -100 | 1172 | 712 | -100 | -100 | -100 |
| 13 | JHALAWAR | -100 | 674 | -100 | -100 | -100 | -100 | -100 | -100 | -100 | -100 |
| 14 | JHUNJHUNUN | -98 | 1194 | -100 | -100 | -100 | 87 | -50 | -100 | -100 | -100 |
| 15 | KARAULI | -80 | 2220 | -100 | -100 | -100 | 167 | -100 | -100 | -100 | -100 |
| 16 | KOTA | -100 | 525 | -100 | -100 | -100 | -100 | -100 | -100 | -100 | -100 |
| 17 | PRATAPGARH | -70 | 518 | -100 | -100 | -100 | -100 | -100 | -100 | -100 | -100 |
| 18 | RAJSMAND | 8 | 351 | -100 | -100 | -100 | -100 | -100 | -100 | -100 | -100 |
| 19 | SIKAR | -100 | 1275 | -100 | -100 | -100 | 177 | 379 | -100 | -100 | -100 |
| 20 | SIROHI | -5 | -87 | -100 | -100 | -100 | -100 | -100 | -100 | -100 | -100 |
| 21 | SWAIMADHOPUR | -100 | 1505 | -100 | -100 | -100 | -37 | -100 | -100 | -100 | -100 |
| 22 | TONK | -81 | 844 | -100 | -100 | -100 | -81 | -100 | -100 | -100 | -100 |
| 23 | UDAIPUR | -73 | 463 | -100 | -100 | -100 | -100 | -100 | -100 | -100 | -100 |
| | SUBDIVISION : WEST MADHYA PRADESH | | | | | | | | | | |
| | STATE : MADHYA PRADESH | | | | | | | | | | |
| 1 | AGAR-MALWA | -81 | 1387 | -100 | -100 | -100 | -100 | -100 | -100 | -100 | -100 |
| 2 | ALIRAJPUR | 9 | 822 | -100 | -100 | -100 | -100 | -100 | -100 | -100 | -100 |
| 3 | ASHOKNAGAR | -95 | 609 | -100 | -100 | -100 | -100 | -100 | -100 | -100 | -100 |
| 4 | BARWANI | -15 | 618 | -97 | -100 | -100 | -100 | -100 | -100 | -100 | -100 |
| 5 | BETUL | -100 | 375 | 192 | -100 | -100 | -100 | -100 | -100 | -100 | -100 |
| 6 | BHIND | -37 | 1074 | -100 | -100 | -100 | -100 | -100 | -100 | -100 | -100 |
| 7 | BHOPAL | -98 | 795 | -100 | -100 | -100 | -100 | -100 | -100 | -100 | -100 |
| 8 | BURHANPUR | -100 | 838 | -70 | -100 | -100 | -100 | -100 | -100 | -100 | -100 |
| 9 | DATIA | -100 | 632 | -61 | -100 | -100 | -100 | -100 | -100 | -100 | -100 |
| 10 | DEWAS | -73 | 794 | -100 | -100 | -100 | -100 | -100 | -100 | -100 | -100 |
| 11 | DHAR | -46 | 446 | -97 | -99 | -100 | -100 | -100 | -100 | -100 | -100 |
| 12 | GUNA | -100 | 408 | -100 | -100 | -100 | -100 | -100 | -100 | -100 | -100 |
| 13 | GWALIOR | -81 | 654 | -34 | -100 | -100 | -100 | -100 | -100 | -100 | -100 |
| 14 | HARDA | -100 | 682 | -100 | -100 | -100 | -100 | -100 | -100 | -100 | -100 |
| 15 | INDORE | -8 | 596 | -100 | -100 | -100 | -100 | -100 | -100 | -100 | -100 |
| 16 | JHABUA | -37 | 388 | -100 | -100 | -100 | -100 | -100 | -100 | -100 | -100 |
| 17 | KHANDWA | -78 | 721 | -100 | -100 | -100 | -100 | -100 | -100 | -100 | -100 |
| 18 | KHARGONE | -44 | 547 | -58 | -100 | -100 | -100 | -100 | -100 | -100 | -100 |
| 19 | MANDSAUR | -61 | 465 | -100 | -100 | -100 | -100 | -100 | -100 | -100 | -100 |
| 20 | MORENA | -100 | 1329 | -100 | -100 | -100 | -100 | -100 | -100 | -100 | -100 |
| 21 | NARMADAPURAM | -80 | 727 | -78 | -100 | -100 | -100 | -100 | -100 | -100 | -100 |
| 22 | NIMACH | -64 | 913 | -100 | -100 | -100 | -100 | -100 | -100 | -100 | -100 |
| 23 | RAISEN | -42 | 744 | -92 | -100 | -100 | -100 | -100 | -100 | -100 | -100 |
| 24 | RAJGARH | -100 | 1112 | -100 | -100 | -100 | -100 | -100 | -100 | -100 | -100 |
| 25 | RATLAM | -72 | 501 | -100 | -100 | -100 | 300 | -100 | -100 | -100 | -100 |
| 26 | SEHORE | -100 | 785 | -100 | -100 | -100 | -100 | -100 | -100 | -100 | -100 |
| 27 | SHAJAPUR | -100 | 1468 | -100 | -100 | -100 | -100 | -100 | -100 | -100 | -100 |
| 28 | SHEOPUR | -100 | 1369 | -100 | -100 | -100 | 1150 | -65 | -100 | -100 | -100 |
| 29 | SHIVPURI | -100 | 968 | -85 | -100 | -100 | -100 | -100 | -100 | -100 | -100 |
| 30 | UJJAIN | -25 | 724 | -100 | -100 | -100 | -100 | -100 | -100 | -100 | -100 |

| # | Name | | | | | | | | | |
|---|---|---|---|---|---|---|---|---|---|---|
| 31 | VIDESHA | -86 | 1295 | -100 | -100 | -100 | -100 | -100 | -100 | -100 | -100 |
| | SUBDIVISION : EAST MADHYA PRADESH | | | | | | | | | |
| 1 | ANUPPUR | -12 | 214 | 32 | -100 | -100 | -100 | -100 | -100 | -100 | -100 |
| 2 | BALAGHAT | -35 | 338 | 73 | -100 | -100 | -100 | -100 | -100 | -100 | -100 |
| 3 | CHHATARPUR | 142 | 678 | -100 | -100 | -100 | -100 | -100 | -100 | -100 | -100 |
| 4 | CHHINDWARA | -84 | 35 | 333 | -100 | -100 | -100 | -100 | -100 | -100 | -100 |
| 5 | DAMOH | 25 | 705 | -100 | -100 | -100 | -100 | -100 | -100 | -100 | -100 |
| 6 | DINDORI | 26 | 570 | 26 | -100 | -100 | -100 | -100 | -100 | -100 | -100 |
| 7 | JABALPUR | 17 | 941 | -100 | -100 | -100 | -100 | -100 | -100 | -100 | -100 |
| 8 | KATNI | -43 | 388 | -96 | -100 | -100 | -100 | -100 | -100 | -100 | -100 |
| 9 | MANDLA | 11 | 467 | 150 | -100 | -100 | -100 | -100 | -100 | -100 | -100 |
| 10 | NARSHIMAPURA | 16 | 855 | -100 | -100 | -100 | -100 | -100 | -100 | -100 | -100 |
| 11 | NIWARI | -66 | 520 | -100 | -100 | -100 | -100 | -100 | -100 | -100 | -100 |
| 12 | PANNA | 192 | 678 | -100 | -100 | -100 | -100 | -100 | -100 | -100 | -100 |
| 13 | REWA | -22 | 727 | -100 | -100 | -100 | -100 | -100 | -100 | -100 | -100 |
| 14 | SAGAR | -60 | 818 | -100 | -100 | -100 | -100 | -100 | -100 | -100 | -100 |
| 15 | SATNA | -66 | 533 | -100 | -100 | -100 | -100 | -100 | -100 | -100 | -100 |
| 16 | SEONI | -76 | 144 | 354 | -100 | -100 | -100 | -100 | -100 | -100 | -100 |
| 17 | SHAHDOL | -23 | 322 | -11 | -100 | -100 | -100 | -100 | -100 | -100 | -100 |
| 18 | SIDHI | 51 | 679 | -89 | -100 | -100 | -100 | -100 | -100 | -100 | -100 |
| 19 | SINGRAULI | 51 | 345 | -89 | -100 | -100 | -100 | -100 | -100 | -100 | -100 |
| 20 | TIKAMGARH | 8 | 387 | -96 | -100 | -100 | -100 | -100 | -100 | -100 | -100 |
| 21 | UMARIA | 31 | 355 | -100 | -100 | -100 | -100 | -100 | -100 | -100 | -100 |
| | SUBDIVISION : GUJARAT REGION | | | | | | | | | |
| | STATE : GUJARAT | | | | | | | | | |
| 1 | AHMADABAD | -100 | 34 | -100 | -100 | -100 | -100 | -100 | -100 | -100 | -100 |
| 2 | ANAND | -100 | 775 | -100 | -100 | -100 | -100 | -100 | -100 | -100 | -100 |
| 3 | ARAVALLI | -89 | 282 | -92 | -100 | -100 | -100 | -100 | -100 | -100 | -100 |
| 4 | BANASKANTHA | -100 | -89 | -100 | -100 | -100 | -100 | -100 | -100 | -100 | -100 |
| 5 | BHARUCH | -66 | 499 | -100 | -100 | -100 | -100 | -100 | -100 | -100 | -100 |
| 6 | CHHOTA UDEPUR | -41 | 335 | -100 | -100 | -100 | -100 | -100 | -100 | -100 | -100 |
| 7 | DAHOD | -78 | 413 | -100 | -100 | -100 | -100 | -100 | -100 | -100 | -100 |
| 8 | DANGS | -35 | 143 | 132 | -15 | -100 | -100 | -100 | -100 | -100 | -100 |
| 9 | GANDHINAGAR | -100 | 233 | -100 | -100 | -100 | -100 | -100 | -100 | -100 | -100 |
| 10 | KHERA | -96 | 591 | -100 | -100 | -100 | -100 | -100 | -100 | -100 | -100 |
| 11 | MAHESANA | -100 | -1 | -100 | -100 | -100 | -100 | -100 | -100 | -100 | -100 |
| 12 | MAHISAGAR | -95 | 827 | -100 | -100 | -100 | -100 | -100 | -100 | -100 | -100 |
| 13 | NARMADA | -45 | 350 | -50 | 1 | -100 | -100 | -100 | -100 | -100 | -100 |
| 14 | NAVSARI | -79 | 172 | 56 | -72 | -100 | -100 | -100 | -100 | -100 | -100 |
| 15 | PANCHMAHAL | -74 | 1378 | -100 | -100 | -100 | -100 | -100 | -100 | -100 | -100 |
| 16 | PATAN | -100 | -100 | -100 | -100 | -100 | -100 | -100 | -100 | -100 | -100 |
| 17 | SABAR KANTHA | -100 | 73 | -100 | -100 | -100 | -100 | -100 | -100 | -100 | -100 |
| 18 | SURAT | -72 | 454 | 431 | -100 | -100 | -100 | -100 | -100 | -100 | -100 |
| 19 | TAPI | -88 | 202 | 892 | -100 | -100 | -100 | -100 | -100 | -100 | -100 |
| 20 | VADODARA | -58 | 549 | -100 | -100 | -100 | -100 | -100 | -100 | -100 | -100 |
| 21 | VALSAD | -57 | 64 | 148 | 480 | -100 | -100 | -100 | -100 | -100 | -100 |
| | STATE : DADRA & NAGAR HAVELI AND DAMAN & DIU (UT) | | | | | | | | | |
| 1 | DADAR & NAGAR HAVELI | -84 | 199 | 417 | 304 | -100 | | | | | |
| 2 | DAMAN | -91 | 74 | -33 | 167 | -100 | -100 | -100 | -100 | -100 | -100 |
| | SUBDIVISION : SAURASHTRA & KUTCH | | | | | | | | | |
| 1 | AMRELI | -84 | 139 | -100 | -100 | -100 | -100 | -100 | -100 | -100 | -100 |
| 2 | BHAVNAGAR | -72 | 221 | -100 | -100 | -100 | -100 | -100 | -100 | -100 | -100 |
| 3 | BOTAD | -92 | -25 | -100 | -100 | -100 | -100 | -100 | -100 | -100 | -100 |

| | | | | | | | | | | |
|---|---|---|---|---|---|---|---|---|---|---|
| 4 | DEVBHOOMI DWARKA | -100 | 56 | -100 | -100 | -100 | -100 | -100 | -100 | -100 | -100 |
| 5 | DIU | -100 | 227 | -100 | -100 | -100 | -100 | -100 | -100 | -100 | -100 |
| 6 | GIR SOMNATH | -100 | 294 | -41 | -100 | -100 | -100 | -100 | -100 | -100 | -100 |
| 7 | JAMNAGAR | -76 | -68 | -100 | -100 | -100 | -100 | -100 | -100 | -100 | -100 |
| 8 | JUNAGARH | -86 | 8 | -100 | -100 | -100 | -100 | -100 | -100 | -100 | -100 |
| 9 | KACHCHH | -71 | -100 | -100 | -100 | -100 | -67 | -100 | -100 | -100 | -100 |
| 10 | MORBI | -100 | -100 | -100 | -100 | -100 | -100 | -100 | -100 | -100 | -100 |
| 11 | PORBANDAR | -52 | -83 | -100 | -100 | -100 | -100 | -100 | -100 | -100 | -100 |
| 12 | RAJKOT | -95 | -15 | -87 | -100 | -100 | -100 | -100 | -100 | -100 | -100 |
| 13 | SURENDRANAGAR | -97 | -100 | -100 | -100 | -100 | -100 | -100 | -100 | -100 | -100 |
| | SUBDIVISION : KONKAN & GOA | | | | | | | | | | |
| | STATE : GOA | | | | | | | | | | |
| 1 | NORTH GOA | -93 | -48 | 131 | 6 | -100 | -99 | -100 | -100 | 109 | -100 |
| 2 | SOUTH GOA | -67 | -61 | -37 | 23 | -100 | -30 | -100 | -100 | 272 | -100 |
| | STATE : MAHARASHTRA | | | | | | | | | | |
| 1 | MUMBAI CITY | -37 | 232 | -27 | -100 | -100 | -100 | -100 | -100 | -100 | -100 |
| 2 | PALGHAR | -59 | 143 | 288 | -57 | -100 | -100 | -100 | -100 | -100 | -100 |
| 3 | RAIGARH | -58 | 217 | 143 | -5 | -100 | -100 | -100 | -100 | -100 | -100 |
| 4 | RATNAGIRI | -68 | 66 | 152 | 37 | -100 | -100 | -100 | -100 | -100 | -100 |
| 5 | SINDHUDURG | -24 | 48 | 236 | -61 | -100 | -94 | -100 | -100 | 495 | -100 |
| 6 | SUBURBAN MUMBAI | -73 | 362 | 652 | 34 | -100 | -100 | -100 | -100 | -100 | -100 |
| 7 | THANE | -47 | 236 | 506 | -54 | -100 | -100 | -100 | -100 | -100 | -100 |
| | SUBDIVISION : MADHYA MAHARASHTRA | | | | | | | | | | |
| 1 | AHMADNAGAR | -50 | 157 | 477 | 294 | -100 | -100 | -100 | -100 | -100 | -100 |
| 2 | DHULE | -15 | 468 | -12 | -100 | -100 | -100 | -100 | -100 | -100 | -100 |
| 3 | JALGAON | -90 | 409 | 216 | 61 | -100 | -100 | -100 | -100 | -100 | -100 |
| 4 | KOLHAPUR | -26 | 156 | 370 | 114 | -100 | -100 | -100 | -100 | 181 | -100 |
| 5 | NANDURBAR | -90 | 303 | 288 | -100 | -100 | -100 | -100 | -100 | -100 | -100 |
| 6 | NASIK | -21 | 112 | 238 | 281 | -100 | -100 | -100 | -100 | -100 | -64 |
| 7 | PUNE | -54 | 51 | 460 | 115 | -100 | -100 | -100 | -100 | -100 | -100 |
| 8 | SANGLI | -25 | 116 | 127 | 150 | -100 | -100 | -100 | -100 | -84 | 43 |
| 9 | SATARA | -50 | 156 | 549 | -20 | -100 | -100 | -100 | -100 | -82 | -100 |
| 10 | SOLAPUR | -67 | 92 | 249 | 124 | -100 | -100 | -100 | -100 | -100 | -100 |
| | SUBDIVISION : MARATHWADA | | | | | | | | | | |
| 1 | AURANGABAD | -87 | 256 | 467 | 47 | -100 | -100 | -100 | -100 | -100 | -100 |
| 2 | BID | -67 | 137 | 373 | 182 | -100 | -100 | -100 | -100 | -100 | -100 |
| 3 | HINGOLI | -82 | 282 | 89 | -26 | -100 | -100 | -100 | -100 | -100 | -100 |
| 4 | JALNA | -45 | 70 | 324 | -37 | -100 | -100 | -100 | -100 | -100 | -100 |
| 5 | LATUR | -71 | 200 | 184 | 84 | -100 | -100 | -100 | -100 | -100 | -100 |
| 6 | NANDED | -71 | 161 | 117 | -79 | -100 | -100 | -100 | -100 | -100 | -100 |
| 7 | OSMANABAD | -77 | 137 | 334 | 239 | -100 | -100 | -100 | -100 | -100 | -100 |
| 8 | PARBHANI | -47 | 98 | 303 | -5 | -100 | -100 | -100 | -100 | -100 | -100 |
| | SUBDIVISION : VIDARBHA | | | | | | | | | | |
| 1 | AKOLA | -75 | 388 | 113 | -76 | -100 | -100 | -100 | -100 | -100 | -100 |
| 2 | AMARAVATI | -89 | 305 | 137 | -99 | -100 | -100 | -100 | -100 | -100 | -100 |
| 3 | BHANDARA | 4 | 501 | 30 | -100 | -100 | -100 | -100 | -100 | -100 | -100 |
| 4 | BULDHANA | -89 | 187 | 374 | 40 | -100 | -100 | -100 | -100 | -100 | -100 |
| 5 | CHANDRAPUR | -92 | -10 | 103 | -89 | -100 | -100 | -100 | -100 | -100 | -100 |
| 6 | GADCHIROLI | -20 | 76 | 51 | -94 | -100 | -100 | -100 | -100 | -100 | -100 |
| 7 | GONDIYA | 26 | 366 | 62 | -98 | -99 | -100 | -100 | -100 | -100 | -100 |
| 8 | NAGPUR | -85 | 276 | 372 | -98 | -99 | -100 | -100 | -100 | -100 | -100 |
| 9 | WARDHA | -90 | 204 | 196 | -87 | -99 | -100 | -100 | -100 | -100 | -100 |
| 10 | WASHIM | -98 | 377 | 133 | -44 | -100 | -100 | -100 | -100 | -100 | -100 |
| 11 | YAVATMAL | -76 | 350 | 141 | -98 | -99 | -100 | -100 | -100 | -100 | -100 |

| | SUBDIVISION : CHHATTISGARH | | | | | | | | | |
|---|---|---|---|---|---|---|---|---|---|---|
| 1 | BALARAMPUR | 246 | 154 | 517 | -100 | -100 | -100 | -100 | -100 | -100 | -100 |
| 2 | BALOD | -21 | 116 | 42 | -100 | -100 | -100 | -100 | -100 | -100 | -100 |
| 3 | BALODA BAZAR | 111 | 99 | 189 | -100 | -100 | -100 | -100 | -100 | -100 | -100 |
| 4 | BASTAR | 62 | -27 | 217 | -93 | -100 | -100 | -100 | -100 | -100 | -100 |
| 5 | BEMETARA | -24 | 6 | 243 | -100 | -100 | -100 | -100 | -100 | -100 | -100 |
| 6 | BIJAPUR | -30 | 42 | 19 | -53 | -100 | -100 | -100 | -100 | -100 | -100 |
| 7 | BILASPUR | 80 | 113 | 105 | -100 | -100 | -100 | -100 | -100 | -100 | -100 |
| 8 | DANTEWARA | -52 | 31 | -31 | -65 | -100 | -100 | -100 | -100 | -100 | -100 |
| 9 | DHAMTARI | -11 | 167 | 26 | -100 | -100 | -100 | -100 | -100 | -100 | -100 |
| 10 | DURG | -14 | 282 | 518 | -100 | -100 | -100 | -100 | -100 | -100 | -100 |
| 11 | GARIABAND | 37 | 53 | 139 | -96 | -100 | -100 | -100 | -100 | -100 | -100 |
| 12 | JANJGIR_CHAMPA | 101 | 74 | 22 | -100 | -100 | -100 | -100 | -100 | -100 | -100 |
| 13 | JASHPUR | 100 | 65 | 415 | -100 | -100 | -100 | -100 | -100 | -100 | -100 |
| 14 | KABIRDHAM | -44 | 172 | 167 | -100 | -100 | -100 | -100 | -100 | -100 | -100 |
| 15 | KANKER | -31 | 173 | 42 | -97 | -100 | -100 | -100 | -100 | -100 | -100 |
| 16 | KONDAGAON | -39 | 9 | 137 | -94 | -100 | -100 | -100 | -100 | -100 | -100 |
| 17 | KORBA | 68 | 36 | 134 | -100 | -100 | -100 | -100 | -100 | -100 | -100 |
| 18 | KOREA | -20 | 261 | 15 | -100 | -100 | -100 | -100 | -100 | -100 | -100 |
| 19 | MAHASAMUND | 41 | -23 | 40 | -100 | -100 | -100 | -100 | -100 | -100 | -100 |
| 20 | MUNGELI | 151 | 305 | 330 | -100 | -100 | -100 | -100 | -100 | -100 | -100 |
| 21 | NARAYANPUR | -36 | -15 | 136 | -50 | -100 | -100 | -100 | -100 | -100 | -100 |
| 22 | RAIGARH | 169 | -2 | 97 | -100 | -100 | -100 | -100 | -100 | -100 | -100 |
| 23 | RAIPUR | 82 | 122 | 149 | -100 | -100 | -100 | -100 | -100 | -100 | -100 |
| 24 | RAJNANDGAON | -1 | 114 | 118 | -100 | -100 | -100 | -100 | -100 | -100 | -100 |
| 25 | SUKMA | 53 | -47 | -13 | -88 | -100 | -100 | -100 | -100 | -100 | -100 |
| 26 | SURAJPUR | 80 | 44 | 275 | -100 | -100 | -100 | -100 | -100 | -100 | -100 |
| 27 | SURGUJA | 24 | -6 | 238 | -100 | -100 | -100 | -100 | -100 | -100 | -100 |
| | SUBDIVISION : COASTAL AP and YANAM | | | | | | | | | |
| 1 | YANAM | -59 | 53 | 249 | -94 | -100 | -45 | -100 | 24 | -91 | -100 |
| | STATE : ANDHRA PRADESH | | | | | | | | | |
| 1 | EAST GODAVARI | -44 | 115 | 108 | -97 | -99 | -73 | -100 | -60 | -88 | -97 |
| 2 | GUNTUR | 47 | 257 | 272 | -95 | -79 | -61 | -87 | -73 | -59 | -100 |
| 3 | KRISHNA | -34 | 167 | 128 | -82 | -89 | -74 | -92 | -60 | -78 | -97 |
| 4 | PRAKASAM | 72 | 197 | -26 | -75 | -52 | -90 | -18 | -78 | -85 | -100 |
| 5 | SPSR NELLORE | -26 | 34 | -74 | -85 | -16 | -69 | 62 | -40 | -25 | -92 |
| 6 | SRIKAKULAM | 55 | 99 | -56 | -98 | -100 | -100 | -100 | -99 | -74 | -100 |
| 7 | VISHAKHAPATNAM | -6 | 127 | 26 | -92 | -99 | -84 | -100 | -99 | 59 | 103 |
| 8 | VIZIANAGARAM | 44 | 210 | 31 | -100 | -100 | -95 | -100 | -100 | -18 | -100 |
| 9 | WEST GODAVARI | -39 | 159 | 128 | -66 | -99 | -75 | -100 | -53 | -73 | -30 |
| | SUBDIVISION : TELANGANA | | | | | | | | | |
| 1 | ADILABAD | -70 | 182 | 152 | -95 | -100 | -100 | -100 | -100 | -100 | -100 |
| 2 | B. KOTHAGUDEM | -42 | 173 | 114 | -79 | -100 | -100 | -100 | -97 | -100 | -100 |
| 3 | HANUMAKONDA | -50 | 82 | 65 | 142 | -100 | -100 | -100 | -100 | -100 | -100 |
| 4 | HYDERABAD | -38 | 120 | 136 | -100 | -98 | -100 | -100 | -100 | -100 | -100 |
| 5 | J. BHUPALPALLY | -85 | 11 | 52 | -100 | -100 | -100 | -100 | -100 | -100 | -100 |
| 6 | JAGTIAL | -82 | 68 | -26 | -95 | -100 | -100 | -100 | -100 | -100 | -100 |
| 7 | JANGAON | 38 | 206 | 30 | 157 | -100 | -100 | -100 | -100 | 27 | -100 |
| 8 | JOGULAMBA GADWAL | 76 | 10 | 86 | -89 | -98 | -99 | -100 | -100 | -100 | -100 |
| 9 | KAMAREDDY | -58 | 80 | 138 | -47 | -99 | -100 | -100 | -100 | -100 | -100 |
| 10 | KARIMNAGAR | -57 | 125 | 224 | -76 | -99 | -100 | -100 | -100 | -100 | -100 |
| 11 | KHAMMAM | -8 | 72 | 51 | -95 | -100 | -89 | -100 | -56 | -96 | -100 |
| 12 | KUMARAM BHEEM | 47 | 94 | 79 | -70 | -100 | -100 | -99 | -100 | -100 | -100 |

| 13 | M. MALKAJGIRI | -54 | 132 | 136 | -53 | -100 | -100 | -100 | -100 | -100 | -100 |
|----|---------------|-----|-----|-----|-----|------|------|------|------|------|------|
| 14 | MAHABUBABAD | 21 | 43 | 97 | -26 | -100 | -78 | -100 | -100 | -45 | -100 |
| 15 | MAHABUBNAGAR | 129 | 247 | 137 | 52 | -94 | -100 | -100 | -97 | -100 | -100 |
| 16 | MANCHERIAL | -42 | 39 | -5 | -100 | -99 | -100 | -100 | -100 | -100 | -100 |
| 17 | MEDAK | -66 | 61 | 124 | -97 | -100 | -100 | -100 | -100 | -100 | -100 |
| 18 | MULUGU | 10 | 65 | 230 | -85 | -100 | -100 | -100 | -100 | -100 | -100 |
| 19 | NAGARKURNOOL | 57 | 135 | 55 | -69 | -95 | -100 | -100 | -96 | -100 | -100 |
| 20 | NALGONDA | 72 | 22 | 192 | -93 | -96 | -97 | -100 | -85 | -77 | -100 |
| 21 | NARAYANPET | 54 | 248 | -25 | 48 | -100 | -100 | -100 | -100 | -100 | -100 |
| 22 | NIRMAL | -77 | 98 | 38 | -85 | -100 | -100 | -100 | -100 | -100 | -100 |
| 23 | NIZAMABAD | -92 | 90 | 18 | -97 | -100 | -100 | -100 | -100 | -100 | -100 |
| 24 | PEDDAPALLE | -78 | 115 | 83 | -98 | -100 | -100 | -100 | -100 | -100 | -100 |
| 25 | RAJANNA SIRCILLA | -60 | 147 | 145 | -50 | -100 | -100 | -100 | -100 | -100 | -100 |
| 26 | RANGAREDDY | 16 | 66 | 115 | -56 | -92 | -100 | -100 | -98 | -100 | -100 |
| 27 | SANGAREDDY | -76 | 65 | 317 | -70 | -100 | -100 | -100 | -100 | -100 | -100 |
| 28 | SIDDIPET | -51 | 97 | 317 | -89 | -100 | -100 | -100 | -98 | -100 | -100 |
| 29 | SURYAPET | 61 | 39 | 172 | -99 | -100 | -66 | -100 | -62 | -91 | -100 |
| 30 | VIKARABAD | -3 | 275 | 88 | -24 | -99 | -100 | -100 | -95 | -100 | -100 |
| 31 | WANAPARTHY | 13 | 304 | 83 | -62 | -75 | -100 | -100 | -88 | -100 | -32 |
| 32 | WARANGAL | -56 | 167 | 129 | 108 | -100 | -100 | -100 | -100 | -100 | -100 |
| 33 | Y. BHUVANAGIRI | 65 | 6 | -16 | -63 | -94 | -100 | -100 | -100 | -100 | -100 |
| | SUBDIVISION : RAYALASEEMA | | | | | | | | | | |
| 1 | ANANTAPURAMU | 48 | 97 | 346 | -92 | -75 | -35 | -34 | -52 | -52 | -100 |
| 2 | CHITTOOR | -66 | 61 | 35 | -66 | -18 | -64 | 42 | -39 | -50 | -94 |
| 3 | KURNOOL | 95 | 55 | 179 | -71 | -94 | -65 | -98 | -80 | -88 | -100 |
| 4 | YSR DISTRICT | 117 | 86 | 22 | -61 | -63 | -57 | 14 | -47 | -17 | -100 |
| | SUBDIVISION : TN PUDU and KARAIKAL | | | | | | | | | | |
| | STATE : TAMIL NADU | | | | | | | | | | |
| 1 | ARIYALUR | -69 | 4 | -3 | 36 | -60 | -57 | 45 | -100 | -98 | 31 |
| 2 | CHENGALPATTU | -79 | 92 | -57 | -60 | 27 | 24 | 142 | -96 | -90 | -85 |
| 3 | CHENNAI | -27 | 81 | -73 | -45 | 91 | 35 | 67 | -86 | -97 | -94 |
| 4 | COIMBATORE | -66 | 54 | 127 | 7 | -79 | -23 | 294 | -94 | 9 | -56 |
| 5 | CUDDALORE | -58 | -23 | 29 | -42 | -25 | 15 | 199 | -98 | -88 | -31 |
| 6 | DHARAMPURI | -76 | 11 | 255 | -64 | -33 | -27 | 92 | -98 | 24 | -81 |
| 7 | DINDIGUL | -71 | 109 | 109 | -6 | -86 | 35 | 214 | -100 | -92 | -9 |
| 8 | ERODE | -45 | 200 | 295 | 78 | -92 | -39 | 293 | -100 | 83 | -17 |
| 9 | KALLAKURICHI | -91 | 35 | 36 | -63 | -53 | 8 | 108 | -100 | -71 | -64 |
| 10 | KANCHIPURAM | -80 | 246 | -73 | -70 | 63 | -7 | 150 | -95 | -53 | -86 |
| 11 | KANYAKUMARI | -72 | -91 | 159 | -34 | -34 | 61 | -5 | -83 | 21 | 129 |
| 12 | KARUR | -76 | 139 | 6 | 0 | -76 | -12 | 203 | -100 | -98 | 7 |
| 13 | KRISHNAGIRI | -65 | 21 | 177 | -11 | -62 | -21 | 224 | -97 | 32 | -98 |
| 14 | MADURAI | -83 | 156 | 176 | 3 | -92 | 32 | 83 | -100 | -49 | 33 |
| 15 | MAYILADUTHURAI | -91 | -83 | 8 | -57 | -38 | 74 | 369 | -100 | -91 | 23 |
| 16 | NAGAPATTINAM | -8 | 104 | 26 | -82 | -7 | 11 | 18 | -100 | -99 | -13 |
| 17 | NAMAKKAL | -22 | 189 | 182 | 28 | -62 | -36 | 262 | -98 | -80 | -44 |
| 18 | NILGIRI | -50 | -28 | 58 | -50 | -89 | -37 | 58 | -100 | -22 | -46 |
| 19 | PERAMBALUR | -84 | 59 | 38 | -14 | -49 | -28 | 87 | -99 | -86 | -19 |
| 20 | PUDUKKOTTAI | -72 | 40 | 81 | -28 | -92 | -48 | 166 | -99 | -78 | 28 |
| 21 | RAMANATHAPURAM | -61 | 99 | 9 | 0 | -50 | 19 | 11 | -100 | -86 | -30 |
| 22 | RANIPET | -69 | 197 | -66 | -70 | -25 | -26 | 81 | -85 | -47 | -90 |
| 23 | SALEM | -76 | 92 | 115 | -45 | -76 | -28 | 139 | -98 | -33 | -63 |
| 24 | SIVAGANGA | -77 | 100 | 111 | -21 | -90 | 41 | 25 | -100 | -89 | 23 |
| 25 | TENI | -88 | -31 | 237 | -51 | -93 | 26 | 94 | -84 | 101 | 4 |
| 26 | TENKASI | -65 | -86 | 25 | -58 | -45 | 25 | 51 | -7 | -48 | -21 |
| 27 | THANJAVUR | -41 | 91 | 33 | -14 | -84 | -26 | 78 | -100 | -88 | 50 |

| 28 | THIRUVARUR | -13 | 28 | -1 | -64 | -57 | 2 | 22 | -100 | -90 | 27 |
|----|------------|-----|-----|-----|-----|-----|-----|-----|------|-----|-----|
| 29 | TIRUCHIRAPPALLI | -66 | 53 | 63 | -4 | -85 | -40 | 80 | -100 | -90 | 6 |
| 30 | TIRUNELVELI | -94 | -26 | -15 | -53 | -46 | 7 | 5 | -30 | -71 | 58 |
| 31 | TIRUPATTUR | -87 | 99 | 248 | -81 | -69 | -49 | 302 | -98 | -24 | -94 |
| 32 | TIRUPPUR | -62 | 108 | 48 | 42 | -89 | -24 | 400 | -99 | -76 | 24 |
| 33 | TIRUVALLUR | -64 | 153 | -58 | -57 | 36 | -42 | 112 | -80 | -63 | -94 |
| 34 | TIRUVANNAMALAI | -91 | 79 | 48 | -64 | -22 | -30 | 146 | -77 | -28 | -80 |
| 35 | TUTICORIN | -86 | -49 | -22 | -37 | 3 | -9 | 3 | -97 | -25 | 2 |
| 36 | VELLORE | -85 | 25 | 85 | -65 | -47 | -53 | 3 | -92 | -27 | -85 |
| 37 | VILLUPURAM | -90 | 127 | 61 | -84 | -3 | -24 | 146 | -90 | -89 | -72 |
| 38 | VIRUDHUNAGAR | -94 | -8 | 85 | -8 | -20 | 48 | 71 | -100 | 23 | -24 |
| | STATE : PUDUCHERRY (UT) | | | | | | | | | | |
| 1 | KARAIKAL | -96 | -41 | -10 | -52 | 15 | -20 | 100 | -100 | -100 | 5 |
| 2 | PUDUCHERY | -98 | 32 | 21 | -14 | 2 | -31 | 119 | -90 | -48 | -46 |
| | SUBDIVISION : COASTAL KARNATAKA | | | | | | | | | | |
| | STATE : KARNATAKA | | | | | | | | | | |
| 1 | DAKSHIN KANNADA | -51 | -21 | -11 | -23 | -94 | -16 | -44 | -99 | 410 | -93 |
| 2 | UDUPI | -67 | -53 | 2 | -52 | -100 | -40 | -67 | -84 | 537 | -94 |
| 3 | UTTAR KANNADA | -38 | 34 | 35 | -38 | -100 | -91 | -100 | -96 | 20 | -100 |
| | SUBDIVISION : N. I. KARNATAKA | | | | | | | | | | |
| 1 | BAGALKOT | 2 | 116 | 297 | -21 | -100 | -98 | -100 | -100 | -100 | -100 |
| 2 | BELGAUM | -17 | 158 | 191 | 71 | -100 | -100 | -100 | -100 | -4 | -100 |
| 3 | BIDAR | -20 | 80 | 266 | 18 | -99 | -100 | -100 | -100 | -100 | -100 |
| 4 | BIJAPUR | -16 | 74 | 134 | 27 | -100 | -99 | -100 | -100 | -100 | -100 |
| 5 | DHARWAD | 42 | 184 | 55 | 71 | -100 | -100 | -100 | -100 | 126 | -100 |
| 6 | GADAG | 141 | 135 | 61 | -61 | -100 | -100 | -100 | -99 | -100 | -100 |
| 7 | GULBARGA | -23 | 99 | 122 | 58 | -100 | -100 | -100 | -98 | -41 | -100 |
| 8 | HAVERI | 6 | 254 | 89 | 10 | -98 | -95 | -100 | -94 | -38 | -100 |
| 9 | KOPPAL | 142 | 166 | 83 | -36 | -100 | -99 | -96 | -65 | -93 | -100 |
| 10 | RAICHUR | 71 | 84 | 173 | -58 | -100 | -71 | -100 | -99 | -100 | -100 |
| 11 | YADGIR | 19 | 177 | -18 | -46 | -100 | 84 | -100 | -88 | -100 | -100 |
| | SUBDIVISION : S. I. KARNATAKA | | | | | | | | | | |
| 1 | BANGALORE RURAL | -41 | 10 | 304 | -27 | -26 | -81 | 89 | -94 | 54 | -100 |
| 2 | BANGLORE URBAN | -26 | 8 | 474 | 50 | -44 | -50 | 67 | -92 | 2 | -100 |
| 3 | BELLARY | 175 | 156 | 243 | 7 | -99 | -65 | -78 | -22 | -90 | -100 |
| 4 | CHAMARAJANAGAR | -56 | 85 | 364 | 27 | -70 | -86 | 159 | -89 | -100 | -91 |
| 5 | CHIKBALLAPUR | -9 | 11 | 303 | -59 | -39 | -78 | -33 | -62 | -5 | -100 |
| 6 | CHIKMAGALUR | -78 | -33 | 126 | -47 | -100 | -66 | -91 | -65 | 274 | -15 |
| 7 | CHITRADURGA | 75 | 84 | 334 | -63 | -98 | 20 | -83 | -66 | -56 | -100 |
| 8 | DAVANGERE | 121 | 241 | 222 | -24 | -100 | -57 | -93 | -86 | -33 | -100 |
| 9 | HASSAN | -28 | 106 | 412 | 36 | -97 | -41 | -89 | -95 | 268 | -31 |
| 10 | KODAGU | -51 | -18 | 107 | 0 | -97 | -57 | -48 | -93 | 140 | -99 |
| 11 | KOLAR | -75 | -16 | 398 | -52 | -11 | -70 | 56 | -25 | 44 | -98 |
| 12 | MANDHYA | -31 | 28 | 465 | 66 | -26 | -44 | 12 | -91 | 161 | -100 |
| 13 | MYSORE | -31 | 129 | 524 | 118 | -96 | -67 | 91 | -80 | 184 | -100 |
| 14 | RAMNAGAR | -25 | 10 | 565 | 42 | 41 | -58 | 179 | -16 | 256 | -100 |
| 15 | SHIMOGA | -57 | -1 | 53 | -14 | -100 | -63 | -96 | -84 | 148 | -100 |
| 16 | TUMKUR | 1 | 61 | 432 | -57 | -92 | -80 | -76 | -72 | -7 | -100 |
| | SUBDIVISION : KERALA & MAHE | | | | | | | | | | |
| 1 | ALAPPUZHA | -45 | -91 | 12 | -20 | -8 | 22 | 53 | -21 | 76 | -25 |
| 2 | ERNAKULAM | -43 | -92 | -2 | -24 | -13 | 32 | -6 | -99 | 496 | 222 |
| 3 | IDUKKI | -41 | -55 | 136 | 8 | -51 | 37 | 5 | -78 | 168 | -8 |
| 4 | KANNUR | -48 | -91 | -67 | -64 | -74 | 56 | -32 | -89 | -13 | -13 |

| | | | | | | | | | | | |
|---|---|---|---|---|---|---|---|---|---|---|---|
| 5 | KASARGOD | -54 | -84 | -6 | 3 | -89 | 32 | -34 | -100 | 518 | -36 |
| 6 | KOLLAM | -51 | -91 | 42 | -40 | -30 | 53 | -25 | -58 | 0 | 0 |
| 7 | KOTTYAM | -22 | -79 | 10 | -31 | 0 | 70 | 49 | -80 | 242 | 56 |
| 8 | KOZIKOD | -60 | -53 | 30 | -23 | -54 | -21 | 20 | -96 | 176 | -31 |
| 9 | MAHE | -75 | -71 | -79 | -81 | -99 | 68 | 146 | -100 | 264 | -50 |
| 10 | MALAPPURAM | -83 | -46 | -33 | -21 | -73 | 39 | -19 | -83 | 2 | 104 |
| 11 | PALAKKAD | -48 | -41 | -20 | -3 | -91 | -58 | 58 | -82 | 27 | -53 |
| 12 | PATTANAMITTIA | -36 | -40 | 16 | 19 | -31 | 13 | 80 | 44 | 477 | 228 |
| 13 | THIRUVANANTHPURAM | -66 | -92 | 213 | -33 | -1 | 7 | -55 | -93 | -75 | 126 |
| 14 | TRISHUR | -87 | -93 | -23 | -53 | -59 | -53 | -7 | -95 | 145 | 12 |
| 15 | WAYANAD | -54 | 14 | 91 | -53 | -90 | -69 | 22 | -89 | -15 | -86 |
| | SUBDIVISION : LAKSHADWEEP | | | | | | | | | | |
| 1 | LAKSHADWEEP | -51 | -14 | 82 | -37 | -79 | -85 | 122 | -99 | -78 | 26 |

---

## Author Response (AR1)

**Reply to Reviewer 1**

We thank both reviewers for their constructive comments. Below we address each question/comment (blue text). In the revised manuscript we will add more details about our reasoning and the chosen methodology in particular the statistical approach.

This paper examines landslide occurrences along a busy highway in India. The authors generate a landslide inventory and analyze corresponding landslide density with respect to some external factors including precipitation, slope and lithology. Overall, it is a publishable work but there are so many points which are not clear to me at all. I have listed all my comments below line-by-line. My main problem with the paper is that the authors do not clearly introduce their methodology and they do not explain reasoning behind choices they made. Therefore, I had a hard time understanding why/how they did some analyses. Following moderate to major revisions, I believe the paper could be publishable.

Lines 29-34. Please cite the relevant literature.

We added a reference to this a recent article:

Boora, S., & Karakunnel, M. T. (2024). The SDG conundrum in India: navigating economic development and environmental preservation. *International Journal of Environmental Studies*, *81*(2), 961–976. https://doi.org/10.1080/00207233.2024.2323321lWe added a reference to a newspaper article.

https://timesofindia.indiatimes.com/india/Govt-plans-Bharat-Mala-a-5000km-road-network/articleshow/47102122.cms#

Lines 35-37. Please cite the relevant literature.

We added a reference to the article:

Sharma, A. K., Parkash, S., & Roy, T. S. (2014). Response to Uttarakhand disaster 2013. *International Journal of Scientific and Engineering Research*, *5*, 1251-1256.

Line 37. Considering the flow of your text, the last line of this paragraph is a bit off.

We'll change it to "Ensuring accessibility and connectivity during such events is thus of live-saving importance, yet requiring considerable maintenance efforts (Uniyal, 2021)."

Line 51. Please cite those "few" studies, cite at least some of them.

We added two references:

Huat, B. B., & Jamaludin, S. (2005). Evaluation of slope assessment system in predicting landslides along roads underlain by granitic formation. *American Journal of Environmental Sciences*, *1*(2), 90-96.

Ching, J., Liao, H. J., & Lee, J. Y. (2011). Predicting rainfall-induced landslide potential along a mountain road in Taiwan. *Geotechnique*, *61*(2), 153-166.

Lines 61-62. "Limiting the inventory to road blocking landslides was required to cope with the overwhelming number of landslides and to ensure that we account for the landslides that detached most recently." Could you please elaborate this further? How do you define those "recently occurred" landslides (e.g., "the ones that blocked roads in the last five years")? And why do you want to put such a restriction to your inventory?

We define "recently occurred" as having occurred during the preceding monsoon season and the unusually heavy rainfall in the two weeks before our survey. In the revised manuscript we specify the periods of high rainfall as the monsoon season May - Aug and the heavy rainfall during 5-12 October (see attached pdf DISTRICT_RAINFALL…., page 6 Uttarakhand).

We initially tried to attribute the landslides to the latter period by matching our data with high-resolution images available in Google Earth but their temporal resolution for many stretches of the road was insufficient. Moreover, to a large part, historic imagery has subsequently been made unavailable over India.

Line 64. "previous studies" Please cite at least some of them.

We added the citations:

Das, I., Stein, A., Kerle, N., and Dadhwal, V. K.: Landslide susceptibility mapping along road corridors in the Indian Himalayas using Bayesian logistic regression models, Geomorphology, 179, 116–125, https://doi.org/10.1016/j.geomorph.2012.08.004, 2012.

Devkota, K. C., Regmi, A. D., Pourghasemi, H. R., Yoshida, K., Pradhan, B., Ryu, I. C., Dhital, M. R., and Althuwaynee, O. F.: Landslide susceptibility mapping using certainty factor, index of entropy and logistic regression models in GIS and their comparison at Mugling–Narayanghat road section in Nepal Himalaya, Nat. Hazards, 65, 135–165, https://doi.org/10.1007/s11069-012-0347-6, 2013.

Line 65. "in two spatial dimensions" No need to indicate this, please remove.

We would rather stick with our formulation to stress the distinctiveness of our approach.

Line 65. "conceptualizes landslides as a network-attached spatial point pattern" Could you please elaborate this further, what do you mean by this? It seems like this is an important element in your methodology and thus, it deserves a bit more attention. What are the main findings of Baddeley and others and how do you think they contribute to the existing literature?

Landslide susceptibility analysis commonly relies on discretizing the study region into pixels where each pixel indicates either the presence or absence of landslides. Subsequently, techniques such as logistic regression are used to predict the probability of a landslide as a function of a number of predictor variables. A pixel-based logistic regression analysis is approximately equivalent to a Poisson point process model if the landslides are originally stored as point features (Baddeley et al. 2010). We take this approach to modeling landslide susceptibility using Point point process models a step further by conceptualizing landslides as point features that are located on or along-side the road network (Okabe and Sugihara 2012). This means that we do not analyze road-attached landslides as events that occur on a continuous and unbounded plane, but rather as events that are along-side of roads. The term "along-side" indicates a somewhat broad spatial relation, but "implies that the physical unit of

the event [...] has an access point on a network" (Okabe and Sugihara 2012, page 7). In our case, this means that a landslide intersects with the road and blocks it. These intersections are, at the scale of our analysis, represented by point features, i.e. points on a network. Our approach relies on the analysis of the spatial arrangement of the points (the point pattern) along the line, and we hope to reveal important features (e.g. trends in point density) with respect to geo-environmental variables.

Baddeley. M. Berman. N.I. Fisher. A. Hardegen. R.K. Milne. D. Schuhmacher. R. Shah. R. Turner. "Spatial logistic regression and change-of-support in Poisson point processes." Electron. J. Statist. 4 1151 - 1201, 2010. https://doi.org/10.1214/10-EJS581

Okabe, A., & Sugihara, K. (2012). Spatial analysis along networks: statistical and computational methods. John Wiley & Sons.

Line 66. "simplified approach" If you provide more explanation we can also have an idea if this is a simplified approach or not.

The comment above also addresses this point. The term "simplified" is not an appropriate term that we will avoid in a revised version of the manuscript because every model entails simplification and abstraction.

Lines 71-72. "We present our results and discuss uncertainties and potential shortcomings of our approach. We conclude with recommendations for refinement of the approach and further research avenues" No need for these lines, you need to have a discussion, conclusions and interpretations anyway.

We will delete these sentences in a revised version of the manuscript.

Line 72. "Study site" You can convert this heading to something like "study area and data" because in the method section you mention your data sources although, for instance, in figure 1 you already present your geology map. Thus, you can move those sections here.

Thanks for pointing out this issue. We will separate the description of data sources more stringently in a revised version of the manuscript.

Line 75. "1000–1200"mm

We will correct the error in the revised version of the manuscript.

Line 115. Method section: Please first provide a summary of your methodology. There are several elements there. If you systematically present your methodology as step1, step2 and so forth, your readers could follow you more easily. For instance, you start with field work and jump into how you mapped landslides. However, it would be better if you list your steps and then explain what you have done (and what datasets you have used) in each of those steps. This is to say that, please first give us a structure and then we can better understand what you did and why you did. In this regard, you can first describe different landslide categories (Figure 1) and indicate those corresponding periods. We need to know the set of criteria you used to label those landslides.

We thank the reviewer for suggesting a better way of presenting and structuring the methods. In a revised version of the manuscript, we will rewrite these sections so that readers are better able to comprehend the approach and its detailed implementation.

Lines 115-130. It seems like you mapped landslides as points but polygons, right? Please indicate this. Could you also mention where you put those points, to the crown of each landslide? Could you please make a figure showing a close-up view showing examples for each landslide category you mentioned? Please show those examples by providing multi-temporal images, so we can see corresponding conditions in different periods. You are saying you collected some GPS locations during your field work. Then I think you should also show us how you assigned those coordinates to corresponding hillslopes.

The reviewer is right in assuming that the landslides were mapped as points using a handheld GPS rather than as polygons. These points were at locations along the road, where landslides intersected and blocked the road. These points were not assigned to the corresponding hillslopes, rather hillslope conditions upslope of the road were assigned to the road network. For example, we calculated slope with a 210 m bufferzone at the side of the road which is located higher than the road, and mapped the average slope within this zone to the road. As such, this value becomes an attribute of the road which serves as a predictor variable during modeling. In a revised version we will clarify this point better.

As mentioned in the manuscript, unfortunately Google Earth has removed all images from the public archive that we used to categorize the landslides. Notwithstanding, we will try to retrieve high-res imagery to showcase the different landslide conditions in different periods (see next answer).

What is the importance of labelling landslides as (1) new landslide, (2) road-blocking landslide visible before the Sep–Oct 2022 rainfall anomaly, and (3) reactivated landslide. In your model, there is no difference between them, right? Here my main concern is that if you used the accumulated precipitation for the corresponding period. For instance, you have some landslides that occurred before Sep-Oct 2022 and you do not know when they really occurred. And yet, you used accumulated precipitation between Jan-Oct, 2022. Also, you have some new landslides, which have obviously occurred in a different time window compared to the former category but you still use the same period to calculate the cumulative precipitation. The same is also valid for the reactivated landslides. I think you should clearly indicate time spans where you think landslides either occurred or reactivated and then you should calculate the cumulative precipitation for the given temporal windows for different landslides separately .

Btw, I think I mentioned this above already, but it is worth mentioning again that you should better describe your landslide categories and their corresponding time windows. You can simply add a table so we can clearly see what you refer to in each category .

We have now repeated the classification of landslides based on the currently available Google Earth (GE) images. Because a lot of historic images are still missing, we used only landslides that most probably occurred between Jan22 and Oct22, encompassing the entire monsoon season of that year and the heavy rainfall period at the beginning of Oct22. This is the same period, for which we accumulated the rainfall/precipitation of the different rainfall products shown in Fig.3. Landslides that certainly occurred before 2022 and those that were hidden in shadows in the GE images were omitted from the analysis. This reduces the number of landslides in the analysis to 250. Among these are reactivated ones that we highlight in Fig.1.

Line 137. Your readers do not have to know what PPS is, please give the long version of it.

PPS is the name of a numeric class in TopoToolbox to study point processes on river networks which we adopted here to work with road networks. PPS stands for "Point patterns on stream networks" and a reference has been provided. Still, we agree that most readers will not be familiar with it and thus we will provide more detail in a revised version of the manuscript.

Line 137. How did you calculate your landslide density btw? This is your target variable and it needs further explanation? First of all, why did you target landslide density and why not landslide occurrences? In this case, you have a density value depending on your moving kernel. It could be small but I assume you will have a value for a large portion of the study area. This means that even if there is no landslide you will have a value. Am I correct? I am asking these questions because I could not find answers in the manuscript. Could you show us the distribution of density values via a histogram. What is the spatial resolution that you create for your landslide density map? In the result section, you should show us that landslide density map. I guess you calculated landslide density around the road with 210m-wide buffer zone (?) Is this the boundary of your study area anyway?

Apparently, it has not been clear from a description that we model the occurrence of landslides along the road, casting the problem into one spatial dimension, i.e. distance along the road. LS density in Fig. 5 is shown as a 1D kernel density, but it could also be shown with a histogram. The pattern will be the same and depend on bin width, rather than kernel bandwidth. If there is no landslide, and a sufficiently small bandwidth, the density will be zero. In fact, a 15 km bandwidth may be too high and we will consider to reduce the bandwidth (or use a histogram instead).

Line 145. Could you please explain why you chose a log-linear model

The reviewer is referred to Baddeley et al. (2010). Our approach is a spatial logistic regression, where the relation between presence probabilities p and explanatory variables X are controlled by the form of the logistic link function (logit $p = \ln(p/(1-p))$. As pixel size tends to zero, we have $p \rightarrow 0$ and $\ln(p/(1-p)) \rightarrow \ln(p)$. The limiting Poisson intensity is thus a loglinear function of the covariates (adopted from Baddeley et al. 2010, page 1166).

Line 165. You do not need to hypothesize this, so you better cite the relevant literature.

From a strictly statistical standpoint it remains a hypothesis that rainfall influences landslide occurrence. Yet we reformulate the sentence and include references to work that have previously shown this to be true.

Line 166. "accumulated rainfall" Be careful that some of those satellite products do not provide rainfall data but precipitation. And these are different things .

We change "rainfall" to "rainfall/precipitation".

Lines 180-184. Please show this layer in one of your figures.

It is shown in Fig. 1 with a white line with a dashed signature.

Line 197. Please cite the relevant literature and explain how it works.

We will add additional information and references in a revised version of the manuscript.

Line 203. I need further explanation for this section. If you use Poisson regression, then I do not know how you convert it into AUC (You have a continuous array but AUC mainly works for binary conditions). I remember I saw it also before in the literature but you do not either cite the literature nor provide a detailed explanation. Therefore, it is not clear for the moment.

The reviewer confuses Poisson regression and Poisson point process models. Both are actually related if there are possibly more than one landslide at a specific location. However, here we are predicting landslides using the latter, which is most similar to a spatial logistic regression. In this case, we are dealing with a binary variable and thus the AUC is warranted.

Also, I would expect to see a correlation between predicted landslide density and calculated value. There you can assess your prediction performance via some other statistical measures.

This correlation is actually shown in Fig. 7c. Correlations between individual predictors and landslide density are shown in Fig. 6.

Line 227. "AIC" Akaike Information Criterion. Please first explain this in your method section and cite the literature as this is not your own methodology.

We have explained AIC in line 210-211 and provided reference to the original publication (Akaike 1974). Nowhere do we claim that this is our own methodology.

Lines 231-234. This needs to be numerically shown, otherwise you can mention it as a source of uncertainty in your discussion section.

We will show this in a revised version of the manuscript.

Lines 238-241. This is not presented in any figure nor via a table. Where are the outputs of the Bayesian feature rank algorithm? I would like to see how you come up with this conclusion .

Thanks for pointing this out. You are right that, except in the text, there is no information about the Bayesian model outputs (except the posterior densities shown in Fig. 4). We'll provide these as a table in a revised version of the manuscript.

Lines 260-266. This needs to be moved to the discussion section.

We would argue against it because these are still results that we are reporting.

Line 287. "this means that spatial variables characterizing the source area (e.g., hillslope gradient) are projected onto the road" What does this mean?

Regarding the hillslope gradient for example, this means that each road pixel is assigned with the hillslope gradient of the adjoining pixels. To be specific we used the mean hillslope gradient of the 7 nearest uphill pixels.

Lines 288-290. I do not think this is relevant, so please remove this. Even if you worked with thousands of landslides that would not be computationally challenging I would say, no?

This refers to our approach that automatically reduces the sample size substantially compared to methods where the landslides are represented as areal features. We think that this potentially more efficient way of allocating computational resources remains noteworthy.

Line 298. "studies come to different conclusions" But in the following lines you do not mention those different conclusions.

We added a sentence that specifies these different conclusions.

"In a validation study of different gridded rainfall products in the Eastern Himalayas Kumar et al. (2021) report that CHIRPS-2.0 overestimated the monsoon but underestimated annual precipitation."

Lines 324-343. "However, we cannot exclude that small-scale topographic changes due to construction or land use changes" You do not know, you are just speculating because you were not able to capture that signal in your model. Therefore, please rewrite this line and make it smoother instead of saying "we cannot exclude".

We will replace it with: "However, small-scale topographic changes due to construction or land use changes (e.g. abandonment and degradation of agricultural fields and terrace systems) may as well exacerbate road-side slope failures (Jacquet et al., 2015; Mauri et al., 2022)."

Line 357. Please remove footnotes and give citations to online sources.

Changed accordingly

Figure 1. "The highest density of landslides occurs between Rishikesh and Srinagar within lithozone 2 and between Pipalkoti and Joshimath in lithozone 1." Please keep this line for your main text and remove it from here.

done

"For description of the lithozones see Table 2."In lines 190-191, you mention five lithologic units: (1), phyllite and shale (2), quartzite (3), quartzite and igneous rocks (4) and crystalline high grade metamorphic rocks (5). Please indicate the same units in Figure 1.

We add the descriptions in the figure caption instead of pointing to Table 2.

"Note that lithozones 0 and 6 are not crossed by the road and are therefore omitted from the description." Refer to them with their names, please do not use ids.

See answer above. We would rather refrain from adding the unit names to the legend because of the space limitation.

"We subdivided the landslides into new ones, reactivated ones and those that were blocking the road before September 2022." Figure captions are just to describe what you present in that figure, it is not to present what you have done. Please say it in the main text not here.

We think that this piece of information is important to fully understand the pie chart given in the figure and would thus choose to keep it.

"Stars indicate locations of the 1999 Mw 6.6 Chamoli earthquake (Kayal et al., 2003; USGS, 2022) and the 2021 Chamoli rock and ice avalanche" Which stars are showing which events?

This should be clear from the dates that are placed besides the stars. We decided to increase the font size within the figure to improve readability.

Figure 2. This figure is not enough to present how you mapped landslides. I made some comments above.

We hope our answers to the comments above have clarified our approach. We think these images are fully suitable to portray the typical landslides we encountered.

Figure 3. Rainfall or precipitation?

We added the respective information in the text and the caption to this figure.

Figure 4. "Lithozone 1 is missing since the parameter is encapsulated in the intercept." I did not understand what you meant here. Please elaborate this in the main text.

Fitting a generalized linear model (or linear model) including a categorical variable with k levels will return k-1 parameters for this variable, instead of k. This happens to avoid multicollinearity and is achieved by setting one of the level-parameters to zero (a reference category). This finally means that the reference category is represented by the model intercept.

Figure 5. First present your predictive variables and then target variables and then finally show you prediction. What are those blue ticks in panel b? How is it possible that accumulated precipitation shows variation in very small distances? The minimum spatial resolution should be 5 km but somehow I see that it changes even within 1 km. Btw, I do not remember if you mention any downscaling but please clarify that point as well if you did not do that.

We think it is fully appropriate to first show the observation (panels a & b), then the predictor variables and eventually the prediction. The blue ticks in panel b) indicate the landslides.

We resampled the rainfall grids to the spatial resolution of the DEM (30 m) as stated in Lines 176 – 178. The wiggles occur because the road "meanders" through the precipitation field.

Table 1. How did you aggregate these units? I see quite similar lithologic units in different categories.

Because the initial units have many individual lithologies incorporated, an overlap between the aggregated lithozones was inevitable. We tried to separate them as best as possible according to a dominating rock type, i.e. carbonate rocks (1), phyllite and shale (2), quartzite (3), quartzite and igneous rocks (4) and crystalline high grade metamorphic rocks (5). We admit that this involves some degree of uncertainty. We presume however, that reassigning a small number of arguable IDs would not change the overall outcome of our analysis.

We thank the reviewer for the constructive criticism. Our responses are highlighted with blue text below.

The research by Mey and co-authors proposes a nice analysis on the occurrence and modelling of landslides along a highway in India. Such research focus is clearly needed in the context of the Anthropocene and the growing concern of the influence of human activities on Earth surface processes. This research is therefore timely and well within the scope of NHESS.

Having had a look at the extensive review already carried out by a first referee; I can only agree with him/her that the manuscript must be improved quite substantially in several aspects. In addition to the issues evidenced by the first reviewer, especially with respect to the methodological aspects, I would like to stress extra points that I hope would help in improving this work.

We hope that our responses to the first referee's comments address most of the reviewer concerns.

- The introduction is somehow a bit strange as it starts with a focus on the study area, without really providing the broader picture of human-induced problem on landslides, and especially that of road. Besides a need of a broader literature (as stressed by the other referee), on would benefit from a better organization of the introduction structure I think.

We see the point that the reviewer made and would start the introduction along the following lines:

"The global increase and expansion of the human population will inevitably promote the transformation of the natural environment. The construction of roads as a key infrastructural element that accompanies this development, impacts the landscape. Particularly in mountainous regions this may entail for example enhanced hillslope erosion and slope instability (Muenchow et al., 2012), affecting the fast growing montane communities especially in the Andes and the Himalayas. A worldwide surge in the proposals for new roads thus warrants the investigation of the link between road network expansion and landslide exposure (Laurance et al., 2014)."

- Although the case study on that highway in India is a nice one, we miss somehow the reason why this case study could be of special interest for a broader audience. In other words, why, for example, would someone working in the Andes be interested in such a study. Here some work could be needed to improve the introduction and discussion.

We have addressed the link to other similarly affected regions in our addition to the introduction. Besides, our methodological approach based on point pattern analysis on linear networks is innovative and can be applied to other regions. We thus believe that our study has a broader significance beyond our study site.

- As the referee says, knowing about the age of the landslides is quite important. However, the age of the road sections is also something that is important. Recent road sections could be much more damaging/impactful on hillslope instability due to, for example, de-buttressing effect that would be more intense in the recent years that follow a new road cut that along cuts of older roads. However, older roads mean also older outcrops, and therefore potentially increased weathering conditions. Maintenance activities and infrastructures to stabilize/drain the road environments can also vary along the road. Overall, I think that information (discussion) on these issues is needed.

We thank the reviewer for pointing this out. We add to our discussion the following:

"The propensity of landsliding along individual road segments may also be affected by their age. Fresh road cuts might initially suffer more from debuttressing than older ones, whereas the long established road cuts will have experienced longer periods of weathering, potentially affecting their stability. Moreover, remedial measures such as slope enforcement or building of retention walls or artificial drainage can be expected to have a stabilizing effect, although we encountered many of such structures that have failed eventually. We appreciate that such small scale disturbances influence the landslide occurrence, their inclusion in the model, however, would require much more data, which are currently unavailable to us."

- 30 m DEM is used to calculate surface (slope?) gradient. At such a resolution we clearly miss the subtle topographic characteristics that we find along roads. In addition, road cuts would definitely be missed. This issue needs to be clarified in the analyses and/or discussion.

We have recognized this fact in the discussion lines 375-384. We hope that with the answer to the previous comment we have satisfactorily addressed the reviewers concern.